# Phase-Type Variational Autoencoders for Heavy-Tailed Data

**Abdelhakim Ziani** [1 2]   **András Horváth** [2]   **Paolo Ballarini** [1]

## Abstract

Heavy-tailed distributions are ubiquitous in real-world data, where rare but extreme events dominate risk and variability. However, standard Variational Autoencoders (VAEs) employ simple decoder distributions (e.g., Gaussian) that fail to capture heavy-tailed behavior, while existing heavy-tail-aware extensions remain restricted to predefined parametric families whose tail behavior is fixed a priori. We propose the *Phase-Type Variational Autoencoder* (PH-VAE), whose decoder distribution is a latent-conditioned Phase-Type (PH) distribution—defined as the absorption time of a continuous-time Markov chain (CTMC). This formulation composes multiple exponential time scales, yielding a flexible, analytically tractable decoder that adapts its finite-range tail behavior directly from the observed data. Experiments on synthetic and real-world benchmarks demonstrate that PH-VAE accurately approximates diverse heavy-tailed distributions, significantly outperforming Gaussian, Student-t, and extreme-value-based VAE decoders in modeling observed tail behavior and extreme quantiles. In multivariate settings, PH-VAE captures realistic cross-dimensional tail dependence through its shared latent representation. To our knowledge, this is the first work to integrate Phase-Type distributions into deep generative modeling, bridging applied probability and representation learning.

## 1. Introduction

Heavy-tailed distributions are prevalent in real-world phenomena and domains, where rare and extreme events significantly influence system behavior (Vogel et al., 2025; Wu et al., 2025; Jia, 2014; Alstott et al., 2014; Resnick, 2007). In natural language processing (NLP), distributions of spoken and written word frequencies are known to exhibit long right tails (Feng et al., 2022; Berman, 2025; Malmgren et al., 2008). In finance, investment returns or losses often follow power-law behavior (Nolan, 2014), leading to substantial risk from extreme events. Similar patterns arise in queuing systems, internet traffic, and many other fields (Li, 2018; Ramaswami et al., 2013; Chakraborty et al., 2022; Alasmar et al., 2021; Howard et al., 2023; Pisarenko & Rodkin, 2010; Delabays & Tyloo, 2022; Papalexiou et al., 2013). These distributions are characterized by high skewness and heavy tails, and, unlike Gaussian distributions, assign non-negligible probability mass far into the tail (Allouche et al., 2024; Foss et al., 2011), making rare or extreme events substantially more likely. This poses a serious challenge for probabilistic modeling: standard assumptions break down, and models must be expressive enough to accurately capture both the body of the distribution, where most observations concentrate, and the tail, where extremes occur. Failure to represent either region can lead to poor generalization, biased predictions, and dangerous underestimation of risk (Adler et al., 1998).

The Variational Autoencoder (VAE) (Kingma & Welling, 2013) is a probabilistic encoder–decoder model that learns a latent-variable representation by optimizing a variational lower bound on the data likelihood. Typically, both the approximate posterior and the decoder likelihood are chosen to be Gaussian for computational tractability (Doersch, 2016; Liang et al., 2024). While this is an appropriate choice for light-tailed and approximately symmetric data, for heavy-tailed data, it fundamentally limits the model's ability to capture extreme behavior, even after extensive training (Floto et al., 2023). Recent work, such as the Extreme VAE (xVAE) (Zhang et al., 2024), addresses this by introducing a power-law-tailed likelihood. However, this approach remains restricted to a specific tail regime, whereas heavy-tailed phenomena exhibit diverse decay behaviors, including Pareto-, Weibull-, and lognormal-type tails (Clauset et al., 2009; Castillo & Puig, 2023; Sormunen et al., 2024).

To overcome these limitations, Phase-Type (PH) distributions (Neuts, 1975) are proposed as the decoder likelihood. PH distributions (Neuts, 1975) are a flexible family defined as absorption times of finite-state continuous-time Markov chains and are widely used in applied probability and performance modeling (Buchholz et al., 2014; Horváth & Telek,

[1]Université Paris Saclay, Lab. MICS, CentraleSupélec, Gif-sur-Yvette, France [2]Università di Torino, Torino, Italy. Correspondence to: Abdelhakim Ziani <hakim.ziani@centralesupelec.fr>.

*Proceedings of the 43rd International Conference on Machine Learning*, Seoul, South Korea. PMLR 306, 2026. Copyright 2026 by the author(s).

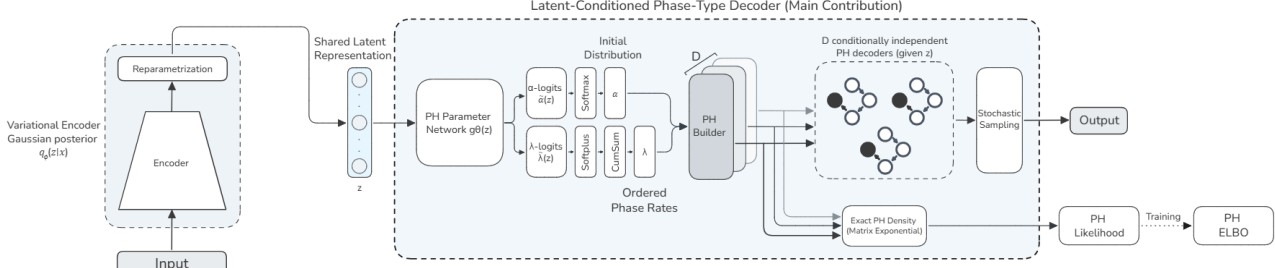

*Figure 1.* Architecture of the Phase-Type Variational Autoencoder (PH-VAE). A shared latent variable $z$ conditions an acyclic Phase-Type decoder, enabling heavy-tailed likelihoods with cross-dimensional dependence.

2024). They can approximate any continuous, positive-valued distribution arbitrarily well (Neuts, 1981), including heavy-tailed families (Horváth & Telek, 2000), while retaining closed-form matrix-exponential expressions for densities and distribution functions. These properties make PH distributions a compelling alternative for modeling complex, skewed, and heavy-tailed data without imposing rigid tail assumptions.

The Phase-Type Variational Autoencoder (PH-VAE) is introduced as a generative model specifically designed to learn from non-negative, multivariate, correlated data with heavy or light tails. PH-VAE follows the classical variational latent formulation but departs fundamentally from standard VAEs in its observation model. Rather than assuming a fixed parametric likelihood, the decoder defines a latent-conditioned stochastic generative mechanism, represented as an absorbing continuous-time Markov process. The resulting Phase-Type likelihood is not a predefined distributional family but a learned composition of exponential time scales, whose shape, including skewness and effective tail behavior, is induced directly from data through the latent space. The key novelty of the proposed approach lies in this flexible decoding mechanism, which generalizes across diverse tail behaviors without committing to a specific extreme-value family. In contrast to previously proposed heavy-tail-aware VAEs (Zhang et al., 2024; Ma et al., 2025), which rely on fixed tail models, PH-VAE learns a distributional structure from data, enabling accurate modeling of both central trends and extreme events. Efficient training is achieved through a PH-based Evidence Lower Bound (ELBO), combining the exact PH log-likelihood with a Gaussian Kullback-Leibler (KL) divergence regularizer. This design unifies deep latent-variable modeling with the expressiveness of stochastic process–based likelihoods.

PH-VAE is evaluated on synthetic and real-world heavy-tailed datasets, including univariate benchmarks with known ground truth and multivariate data exhibiting controlled dependence or real financial correlations. Evaluation focuses on both marginal tail fidelity and cross-dimensional dependence using correlation, rank-based, and tail co-exceedance metrics.

The remainder of the paper is organized as follows. Section 2 reviews background and related work, Section 3 details the proposed model and optimization objective, Section 4 presents the experimental results, and Section 5 concludes the paper.

## 2. Background and Related Work

### 2.1. Variational Autoencoders

Variational Autoencoders (VAEs), introduced by (Kingma & Welling, 2013), are latent-variable generative models that combine neural networks with variational inference. Given an observation $x \in \mathbb{R}^D$, VAEs posit a latent variable $z \in \mathbb{R}^d$ and define a joint distribution $p_\theta(x, z) = p_\theta(x|z)p(z)$, where $p(z)$ is a simple prior (typically standard normal). Learning proceeds by approximating the intractable posterior $p_\theta(z|x)$ with a variational distribution $q_\phi(z|x)$ parameterized by an encoder network. Here, $\phi$ parameterizes the encoder $q_\phi(z \mid x)$, while $\theta$ parameterizes the decoder $p_\theta(x \mid z)$.

**ELBO formulation.** Model parameters are learned by maximizing the evidence lower bound (ELBO), which provides a tractable surrogate to the marginal log-likelihood $\log p_\theta(x)$:

$$\mathcal{L}(\theta, \phi; x) = \mathbb{E}_{q_\phi(z|x)}\big[\log p_\theta(x|z)\big] \\ - \mathrm{KL}[q_\phi(z|x) \, \| \, p(z)]. \quad (1)$$

The reconstruction term encourages fidelity to the data, while the KL divergence regularizes the approximate posterior toward the prior. The expectation is optimized using the reparameterization trick (Kingma & Welling, 2013), enabling efficient gradient-based learning.

**Decoder likelihood and reconstruction.** For continuous data, the decoder distribution $p_\theta(x|z)$ is most commonly

assumed to be Gaussian with diagonal covariance,

$$p_\theta(x|z) = \mathcal{N}\big(x \mid \mu_\theta(z), \mathrm{diag}(\sigma_\theta^2(z))\big),$$

where neural networks parameterize the conditional mean $\mu_\theta(z)$ and variance $\sigma_\theta^2(z)$ (Cemgil et al., 2020). Under this assumption, the expected reconstruction under the decoder satisfies

$$\mathbb{E}_{p_\theta(x|z)}[x] = \mu_\theta(z),$$

and minimizing the negative log-likelihood is equivalent to minimizing a weighted squared-error loss (Kingma et al., 2019; Chen et al., 2017). As a result, reconstructions commonly reported in practice correspond to the decoder mean $\mu_\theta(z)$ rather than samples from $p_\theta(x|z)$.

This Gaussian modeling choice implicitly constrains the conditional distribution $p_\theta(x|z)$ to be light-tailed for every fixed $z$. Consequently, even if the latent variable $z$ is expressive, the decoder can only represent heavy-tailed behavior through variability in the conditional means $\mu_\theta(z)$ across latent space, rather than through genuinely heavy-tailed conditional distributions.

**Limitations.** While VAEs constitute a flexible and elegant probabilistic framework, their expressivity is constrained by modeling assumptions at multiple levels of the architecture. Much prior work has focused on increasing the flexibility of the latent representation, for example via hierarchical latent structures (Klushyn et al., 2019) or normalizing flow-based variational posteriors (Pires & Figueiredo, 2020; Tomczak & Welling, 2018; Kuzina & Tomczak, 2024). In contrast, comparatively less attention has been paid to limitations arising from fixed assumptions on the decoder likelihood.

In particular, when modeling skewed or heavy-tailed data, a Gaussian decoder induces a mismatch between the true tail behavior and the conditional distribution $p_\theta(x|z)$. Because this mismatch results in tail collapse and poor extrapolation, current decoders often fail to capture extreme events, motivating the development of alternative decoder families that can represent heavy-tailed conditional distributions directly.

## 2.2. Beyond Classical VAEs

Several recent studies have highlighted the challenges of modeling real-world heavy-tailed data with VAEs (Lafon et al., 2023) and have proposed modifications to the classical architecture. One notable direction is the t3-VAE model (Kim et al., 2023; Bouayed et al., 2025), which replaces the standard Gaussian components with Student's $t$ distributions in the prior, encoder, and decoder. t3-VAE employs a reformulated training objective based on the $\gamma$-power divergence and introduces a single hyperparameter $\nu$ to control tail heaviness, thereby increasing flexibility in the latent representation. While t3-VAE demonstrates improved performance over Gaussian VAEs, it remains constrained

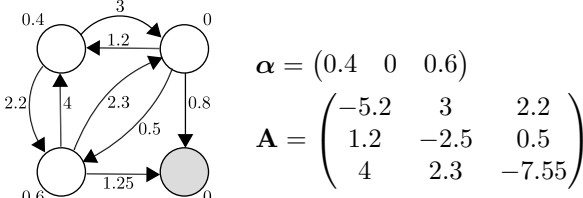

$$\boldsymbol{\alpha} = \begin{pmatrix} 0.4 & 0 & 0.6 \end{pmatrix}$$
$$\mathbf{A} = \begin{pmatrix} -5.2 & 3 & 2.2 \\ 1.2 & -2.5 & 0.5 \\ 4 & 2.3 & -7.55 \end{pmatrix}$$

*Figure 2.* A Phase-Type distribution with three transient states.

by its reliance on a fixed Student's $t$ family with a single degree-of-freedom parameter $\nu$ shared across the prior, encoder, and decoder. As reported in the original studies, model performance is sensitive to the choice of $\nu$, and the associated reparameterization and divergence formulation ($\gamma$-loss) restricts the decoder to a specific power-law tail behavior.

Another related line of work is the Extreme Variational Autoencoder (xVAE) (Zhang et al., 2024), which targets the modeling of spatial extreme events in geophysical data. xVAE introduces max-infinitely divisible (max-id) processes into the VAE framework to better capture rare events. Rather than assuming Gaussian output distributions, xVAE employs exponentially tilted stable distributions in the decoder and is evaluated on a wildland fire plume application. The model shows improved performance over classical approaches such as Proper Orthogonal Decomposition (POD) in capturing marginal and joint tail behavior. However, by relying on a specific class of stable distributions, xVAE still assumes a fixed decoder tail structure, which limits its adaptability across heterogeneous heavy-tailed regimes.

## 2.3. Phase-Type Distributions

A Phase-Type (PH) distribution is a probability distribution on $\mathbb{R}_+$ defined as the time to absorption in a finite-state continuous-time Markov chain (CTMC) consisting of one absorbing state and a finite number of transient states, referred to as phases (Neuts, 1975). PH distributions admit a compact representation through a vector-matrix pair $(\boldsymbol{\alpha}, \mathbf{A})$, where $\boldsymbol{\alpha}$ denotes the initial distribution over transient states and $\mathbf{A}$ is the sub-generator matrix corresponding to transient states. In $\boldsymbol{\alpha}$ and $\mathbf{A}$ the absorbing state is implicit, as transition intensities toward absorption are fully determined by $\mathbf{A}$. For an illustration, see Figure 2.

While Phase-Type distributions are asymptotically light-tailed, they can approximate a wide range of heavy-tailed behaviors arbitrarily well over finite, data-relevant regimes (Horváth & Telek, 2000), which is the focus of this work.

Given $(\boldsymbol{\alpha}, \mathbf{A})$, the probability density function (PDF), cumulative distribution function (CDF), and complementary CDF (CCDF) admit closed-form matrix-exponential expressions:

$$f(x) = \boldsymbol{\alpha} e^{x\mathbf{A}}(-\mathbf{A}\mathbf{1}), \ F(x) = 1 - \boldsymbol{\alpha} e^{x\mathbf{A}}\mathbf{1},$$
$$\bar{F}(x) = \boldsymbol{\alpha} e^{x\mathbf{A}}\mathbf{1}. \tag{2}$$

where $\mathbf{1}$ denotes a column vector of ones. These expressions enable exact and efficient evaluation of likelihoods and tail probabilities, making PH distributions particularly attractive for likelihood-based generative models.

Phase-Type distributions form a highly expressive family on $\mathbb{R}_+$, allowing accurate approximation of a wide range of distributional shapes (asm, 2003). Importantly, PH distributions combine this expressiveness with strong analytical and numerical tractability: key quantities such as densities, tail probabilities, moments, and Laplace transforms are available in closed form through matrix-exponential expressions. These properties make PH distributions well-suited for modeling skewed and heavy-tailed data within deep generative frameworks.

An illustrative example of approximating a heavy-tailed Weibull distribution using a PH distribution is provided in Appendix C. Appendix C further presents a detailed theoretical discussion of Phase-Type distributions, including their Markov chain construction, universality properties, approximations of heavy-tailed distributions, asymptotic considerations, and sampling procedures.

## 3. Phase-Type Variational Autoencoder

### 3.1. Model Overview and Architecture

The Phase-Type Variational Autoencoder (PH-VAE) is a latent-variable generative model designed for multivariate, correlated, positive data with heavy or light tail. Let $x = (x_1, \ldots, x_D) \in \mathbb{R}_+^D$ denote an observation with $D$ dimensions. PH-VAE follows the standard variational autoencoder formulation with a Gaussian latent variable $z \in \mathbb{R}^d$, while replacing the decoder likelihood with a latent-conditioned Phase-Type generative mechanism.

The encoder defines a variational posterior

$$q_\phi(z|x) = \mathcal{N}\left(z \mid \boldsymbol{\mu}_\phi(x), \mathrm{diag}(\boldsymbol{\sigma}_\phi^2(x))\right), \qquad (3)$$

from which latent samples are obtained using the standard reparameterization trick. The prior distribution is chosen as $p(z) = \mathcal{N}(\mathbf{0}, \mathbf{I})$.

### 3.2. Latent-Conditioned Phase-Type Decoder

Conditioned on a latent sample $z$, the decoder defines a multivariate likelihood through dimension-wise Phase-Type distributions. Specifically, PH-VAE assumes conditional independence given $z$:

$$p_\theta(x|z) = \prod_{j=1}^{D} p_\theta(x_j|z), \qquad (4)$$

where each marginal likelihood $p_\theta(x_j|z)$ is modeled as a Phase-Type distribution.

For each dimension $j$, the decoder outputs the representation of a PH distribution $(\boldsymbol{\alpha}_j(z), \mathbf{A}_j(z))$ where $\boldsymbol{\alpha}_j(z) \in \mathbb{R}^m$ is the vector of initial probabilities over $m$ transient states and $\mathbf{A}_j(z) \in \mathbb{R}^{m \times m}$ is a valid sub-generator matrix. To ensure numerical stability and parameter efficiency (Cumani, 1982), each such PH distribution is parameterized as an *acyclic* PH distribution in the *series canonical form*. An acyclic PH distribution has a sub-generator matrix with no cycles in its underlying Markov chain. The class of PH distributions in the series canonical form is distributionally equivalent to the entire acyclic PH class — a dense family of distributions (see Thm. 4.2 on page 84 of (asm, 2003) and C.5 for details) — while requiring only $m$ parameters to specify the $m \times m$ sub-generator matrix, rather than $m^2$ parameters in the general case. The sub-generator in series canonical form is

$$\mathbf{A}_j(z) = \begin{bmatrix} -\lambda_{j,1} & \lambda_{j,1} & 0 & \cdots & 0 \\ 0 & -\lambda_{j,2} & \lambda_{j,2} & \cdots & 0 \\ \vdots & \vdots & \ddots & \ddots & \vdots \\ 0 & 0 & \cdots & -\lambda_{j,m-1} & \lambda_{j,m-1} \\ 0 & 0 & \cdots & 0 & -\lambda_{j,m} \end{bmatrix},$$

with increasing rates $0 < \lambda_{j,1} \leq \cdots \leq \lambda_{j,m}$.

Under this construction, the conditional likelihood for dimension $j$ is

$$p_\theta(x_j|z) = \boldsymbol{\alpha}_j(z) \exp(\mathbf{A}_j(z)x_j)(-\mathbf{A}_j(z)\mathbf{1}). \qquad (5)$$

Although the conditional likelihood factorizes across dimensions (see Equation (4)), statistical dependence between components of $x$ is induced through the shared latent variable $z$. This design allows PH-VAE to capture both marginal behavior and cross-dimensional dependence without explicitly specifying a joint multivariate distribution.

---

**Algorithm 1** Latent-Conditioned Series canonical form Phase-Type constraint setter

---

**Input:** Latent sample $z \in \mathbb{R}^d$, number of phases $m$, dimensions $D$
**Output:** PH parameters $\{(\boldsymbol{\alpha}_j, \mathbf{A}_j)\}_{j=1}^{D}$
$h \leftarrow \mathrm{MLP}_{\mathrm{dec}}(z)$
**for** $j = 1$ **to** $D$ **do**
   $\boldsymbol{\alpha}_j \leftarrow \mathrm{softmax}(W_j^{\boldsymbol{\alpha}}h), \quad \tilde{\boldsymbol{\lambda}}_j \leftarrow \mathrm{softplus}(W_j^{\boldsymbol{\lambda}}h),$
   $\boldsymbol{\lambda}_j \leftarrow \mathrm{cumsum}(\tilde{\boldsymbol{\lambda}}_j)$
   Construct Series canonical form generator $\mathbf{A}_j$ with $(\mathbf{A}_j)_{ii} = -\lambda_{j,i}, (\mathbf{A}_j)_{i,i+1} = \lambda_{j,i}$
**return** $\{(\boldsymbol{\alpha}_j, \mathbf{A}_j)\}_{j=1}^{D}$

---

Algorithm 1 details how the decoder maps a latent sample $z$ to valid Phase-Type parameters. The softmax ensures a proper initial distribution $\boldsymbol{\alpha}_j$, while the cumulative sum of positive rates enforces the required ordering

$0 < \lambda_{j,1} \leq \cdots \leq \lambda_{j,m}$, guaranteeing a valid series canonical form generator matrix.

Unlike common VAE implementations that visualize or generate samples using only the decoder mean, our model generates observations by sampling directly from the conditional likelihood, ensuring that tail behavior is governed by the assumed distribution rather than by moment extrapolation.

### 3.3. Training Objective

Model parameters are learned by maximizing a Phase-Type–based Evidence Lower Bound (ELBO). Given a dataset $\mathcal{D} = \{x^{(i)}\}_{i=1}^{N}$, the objective for a single observation $x \in \mathbb{R}_+^D$ is

$$\mathcal{L}(\theta, \phi; x) = \mathbb{E}_{q_\phi(z|x)}\left[\sum_{j=1}^{D} \log p_\theta(x_j|z)\right]$$
$$- \beta\, D_{\mathrm{KL}}(q_\phi(z|x) \,\|\, p(z)), \tag{6}$$

where $\beta$ controls the strength of the KL regularization, as in the $\beta$-VAE framework. By Equation (4), the reconstruction term decomposes into a sum of log-densities of Phase-Type distributions, with the density given in Equation (5). Each log-density can be computed efficiently and stably via the *uniformization method* (Stewart, 1994). For a generator matrix $\mathbf{A}$ and absorption time $x$, this method computes the matrix exponential based on

$$\exp(Ax) = \sum_{k=0}^{\infty} e^{-\Lambda x}\frac{(\Lambda x)^k}{k!}P^k, \qquad P = I + \frac{A}{\Lambda}, \quad (7)$$

where $\Lambda \geq \max_i(-A_{ii})$ is chosen to ensure $P$ is stochastic, and the infinite sum is truncated to achieve a specified numerical precision.

## 4. Experiments

We evaluate PH-VAE on (i) synthetic 1D heavy-tailed benchmarks with known ground truth, (ii) real-world 1D heavy-tailed datasets, and (iii) multivariate heavy-tailed settings, including both synthetic data with controlled dependence and real financial returns. Across all settings, we compare against Gaussian VAE baselines and heavy-tail-aware decoders (t-VAE, xVAE) under matched architectures and optimization budgets. Ablations on the number of PH phases and additional qualitative diagnostics are reported in Section 4.2.

**Evaluation Metrics.** Our evaluation distinguishes between *marginal tail fidelity* and *multivariate dependence*, since the latter becomes central once moving beyond 1D.

**(1) Tail fidelity in 1D.** For synthetic 1D datasets, where the ground-truth distribution $F$ is known, we quantify tail-shape

recovery using the *conditional tail Kolmogorov-Smirnov distance* ($\mathrm{KS}_{\mathrm{tail}}$). Let $q_{0.95} = F^{-1}(0.95)$. We compare conditional CDFs restricted to the upper tail and renormalized:

$$\mathrm{KS}_{\mathrm{tail}} = \sup_{x \geq q_{0.95}} \left| F_{\mathrm{model}|x>q_{0.95}}(x) - F_{\mathrm{true}|x>q_{0.95}}(x) \right|.$$

We additionally report the *Q99 error*, defined as the relative deviation between the model and true 99th percentiles, which directly probes extreme-value accuracy. Formal definitions and implementation details are provided in Appendix A

**(2) Real-world 1D diagnostics.** For real 1D datasets, the true distribution is unknown, and extreme tails are often affected by discretization and sampling variability. We therefore emphasize qualitative tail diagnostics (log-log CCDF plots) to assess whether models reproduce the empirical decay over several orders of magnitude.

**(3) Multivariate dependence metrics.** For multivariate data, we evaluate whether the model captures cross-dimensional dependence and joint extremes. We report three complementary statistics computed between real data samples and model-generated samples: (i) Frobenius norm of the correlation-matrix error on $\log(1 + x)$ transformed data (the transformation reduces the impact of outliers, allowing the metric to focus on typical feature relationships), (ii) average absolute error in pairwise Kendall's $\tau$ (rank dependence between feature pairs, robust to non-linear monotonic relationships) (Dehling et al., 2017), and (iii) *tail co-exceedance* error (Straetmans et al., 2008).

To define the latter, for a given quantile level $q \in (0, 1)$, let $Q_i(q)$ denote the $q$-th marginal quantile of dimension $i$. The empirical tail co-exceedance probability for a pair $(i, j)$ is defined as

$$\widehat{p}_{i,j}(q) = \frac{1}{N}\sum_{n=1}^{N}\mathbf{1}\left[x_i^{(n)} > Q_i(q) \,\wedge\, x_j^{(n)} > Q_j(q)\right]. \tag{8}$$

which quantifies how often, on average, dataset entries jointly appear to be extreme w.r.t. a pair of dimensions $i$ and $j$. We report the average absolute difference in tail co-exceedance probability between real and generated data:

$$\mathrm{CoExErr}(q) = \frac{2}{D(D-1)} \sum_{1 \leq i < j \leq D} \left|\widehat{p}_{i,j}^{\mathrm{gen}}(q) - \widehat{p}_{i,j}^{\mathrm{real}}(q)\right|. \tag{9}$$

This metric averages absolute pairwise co-exceedance discrepancies across dimension pairs, preventing cancellation effects and isolating whether a shared latent representation induces realistic dependence beyond matching marginals.

### 4.1. Synthetic Heavy-Tailed 1D Distributions

We begin by evaluating PH-VAE on heavy-tailed distributions for which the ground-truth density and tail behavior are



Synthetic Heavy-Tailed Data from Weibull Distribution

- Histogram (Density)
- Weibull PDF (k=0.8, λ=1.0)

(a) True Weibull

VAE generations vs Weibull

- Weibull PDF (k=0.8, λ=1.0)
- Histogram (Density)

(b) VAE

PH-VAE generations vs Weibull

- Histogram (Density)
- Weibull PDF (k=0.8, λ=1.0)

(c) PH-VAE



*Figure 3.* Synthetic Weibull data: true samples, Gaussian VAE generations, and PH-VAE generations.

*Table 1.* Quantitative evaluation of tail reconstruction on synthetic datasets. $KS_{tail}$ is computed on the conditional CDF above the 95th percentile, and Q99 is the relative deviation of the 99th percentile. Across all four distributions, PH-VAE achieves the strongest tail approximation on nearly all datasets and metrics, highlighting the flexibility of the PH decoder relative to fixed-family decoders.

| Model | Weibull ($k = 0.8, \lambda = 1.0$) $KS_{tail}\downarrow$ | Q99 Error $\downarrow$ | Pareto ($\alpha = 2.4, x_m = 1.0$) $KS_{tail}\downarrow$ | Q99 Error $\downarrow$ | Lognormal ($\mu = 0, \sigma = 1.5$) $KS_{tail}\downarrow$ | Q99 Error $\downarrow$ | Burr ($c = 1.5, k = 0.8$) $KS_{tail}\downarrow$ | Q99 Error $\downarrow$ |
|---|---|---|---|---|---|---|---|---|
| VAE | $0.190 \pm 0.012$ | $0.235 \pm 0.003$ | $0.259 \pm 0.005$ | $0.336 \pm 0.002$ | $0.498 \pm 0.007$ | $0.582 \pm 0.002$ | $0.433 \pm 0.002$ | $0.724 \pm 0.003$ |
| t-VAE | $0.225 \pm 0.090$ | $0.878 \pm 0.000$ | $0.233 \pm 0.025$ | $0.632 \pm 0.006$ | $0.650 \pm 0.001$ | $100.783 \pm 1.683$ | $0.518 \pm 0.002$ | $39.845 \pm 1.837$ |
| xVAE | $0.187 \pm 0.008$ | $0.062 \pm 0.005$ | $0.161 \pm 0.002$ | $0.131 \pm 0.008$ | $0.032 \pm 0.005$ | $0.209 \pm 0.020$ | $N/A$ | $1.000 \pm 0.000$ |
| LogNormal-VAE | $0.326 \pm 0.023$ | $2.088 \pm 0.381$ | $0.072 \pm 0.046$ | $0.197 \pm 0.079$ | $0.026 \pm 0.010$ | $0.082 \pm 0.051$ | $0.143 \pm 0.010$ | $0.379 \pm 0.056$ |
| Gamma-VAE | $0.056 \pm 0.004$ | $0.082 \pm 0.012$ | $0.236 \pm 0.100$ | $0.342 \pm 0.030$ | $0.136 \pm 0.048$ | $0.306 \pm 0.050$ | $0.083 \pm 0.015$ | $\mathbf{0.166 \pm 0.087}$ |
| PH-VAE | $\mathbf{0.022 \pm 0.003}$ | $\mathbf{0.010 \pm 0.002}$ | $\mathbf{0.051 \pm 0.003}$ | $\mathbf{0.023 \pm 0.012}$ | $\mathbf{0.020 \pm 0.005}$ | $\mathbf{0.016 \pm 0.009}$ | $\mathbf{0.080 \pm 0.008}$ | $0.599 \pm 0.008$ |

known. This setting allows us to quantify the model's ability to recover both the body and the extreme tail of the distribution. We consider four representative heavy-tailed families: Weibull, Pareto, Lognormal, and Burr, each with 10,000 samples. All models (VAE, t3-VAE, xVAE, and PH-VAE) are trained under identical architectures and optimization settings to ensure fair comparison.

Figure 3 illustrates the qualitative behavior of the models on the Weibull dataset, chosen as a representative example of a non–power-law heavy-tailed distribution. While Gaussian VAE collapses the upper tail and only outputs generations in the body region, PH-VAE exhibits a close visual match to both the body and tail, capturing the curvature of the distribution that the baselines miss.

Table 1 reports $KS_{tail}$ and Q99 error across all four synthetic datasets. PH-VAE achieves the lowest tail discrepancy and quantile error in every case, consistently outperforming competing baselines. On the Burr dataset, the xVAE baseline exhibits tail collapse, leading to a degenerate estimate of the 99th percentile and a constant relative Q99 error of 1.0. These results demonstrate that PH-VAE adapts flexibly to a wide range of heavy-tail behaviors – including exponential-like, lognormal-like, and power-law regimes – while baseline VAEs remain limited by their assumed output families. Despite this expressiveness, the decoder remains lightweight thanks to its compact PH representation, making training computationally efficient.

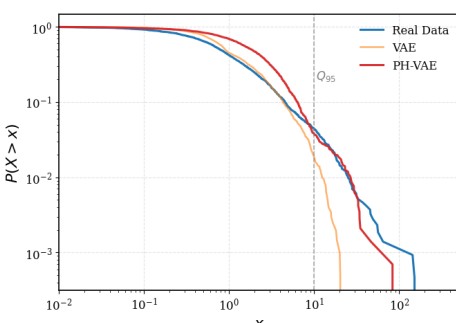

*Figure 4.* Log-log CCDF of Danish Fire Insurance losses for real data, Gaussian VAE, and PH-VAE.

### 4.2. Real-World Heavy-Tailed Univariate Data

Unlike synthetic settings, for real-world datasets, the true data-generating distribution is unknown, and extreme tails are affected by sampling noise and discretization. These experiments therefore, serve as robustness tests, assessing whether PH-VAE can qualitatively reproduce heavy-tailed phenomena that classical Gaussian VAEs systematically fail to capture. We consider two canonical benchmarks: insurance claim sizes from the Danish Fire Insurance dataset (McNeil, 1997; Shongwe & Marambakuyana, 2024) and word-frequency counts from the Google Web Trillion Word Corpus (Brants & Franz, 2006). In both cases, evaluation focuses on qualitative tail behavior using log-log CCDFs.

Figure 4 shows the CCDF of Danish Fire Insurance losses, a standard benchmark in extreme-value analysis where rare but severe events dominate aggregate risk. Figure 5 reports the CCDF for word-frequency counts, which ex-

*Table 2.* Relative tail-risk errors on real-world heavy-tailed datasets. Lower is better. PH-VAE consistently reduces both extreme quantile and CVaR errors compared with the Gaussian VAE baseline.

| Metric | $q$ | Danish Fire | | Word Frequency | |
| --- | --- | --- | --- | --- | --- |
| | | PH-VAE ↓ | VAE ↓ | PH-VAE ↓ | VAE ↓ |
| QErr | 0.950 | $0.130 \pm 0.052$ | $0.423 \pm 0.009$ | $0.065 \pm 0.029$ | $0.435 \pm 0.016$ |
| | 0.990 | $0.456 \pm 0.082$ | $0.654 \pm 0.018$ | $0.096 \pm 0.047$ | $0.515 \pm 0.036$ |
| | 0.995 | $0.334 \pm 0.050$ | $0.702 \pm 0.015$ | $0.070 \pm 0.056$ | $0.558 \pm 0.040$ |
| CVaRErr | 0.950 | $0.263 \pm 0.087$ | $0.677 \pm 0.007$ | $0.219 \pm 0.039$ | $0.636 \pm 0.034$ |
| | 0.990 | $0.200 \pm 0.137$ | $0.809 \pm 0.005$ | $0.332 \pm 0.043$ | $0.720 \pm 0.043$ |
| | 0.995 | $0.192 \pm 0.140$ | $0.854 \pm 0.003$ | $0.402 \pm 0.043$ | $0.757 \pm 0.049$ |

hibit well-known Zipf-like behavior and extreme skewness. Across both datasets, the Gaussian VAE collapses in the tail, severely underestimating rare events, whereas PH-VAE closely follows the empirical CCDF over multiple orders of magnitude, indicating substantially improved modeling of heavy-tailed behavior.

**Ablation.** We analyze the sensitivity of PH-VAE to two key hyperparameters of the decoder: the number of phases $m$, which controls the expressiveness of the resulting Phase-Type distribution, and the KL regularization weight $\beta$ in the ELBO. Results on synthetic Pareto data are reported in Table 3.

Varying the number of phases with fixed $\beta = 1$, we observe that small values of $m$ underfit the tail, leading to higher extreme-quantile errors, while moderate values achieve the best trade-off between tail fidelity and stability. In particular, $m = 10$, used as the default setting throughout the paper, yields the lowest tail co-exceedance error, while slightly larger values marginally improve tail shape at the cost of increased variance. This confirms that increasing the number of phases improves flexibility only up to a point, after which optimization becomes less stable.

Additionally, Table 6 in Section D.3 reports per-epoch time, total training time, and final Negative Log Likelihood (NLL) for different numbers of PH phases. We further study the effect of the KL weight $\beta$ with fixed $m = 10$. Reducing $\beta$ weakens regularization and results in unstable tail behavior and large quantile errors, whereas overly strong regularization degrades extreme-value accuracy. Overall, $\beta = 1$ consistently provides a robust balance between reconstruction accuracy and latent regularization. These results indicate that PH-VAE is not overly sensitive to hyperparameter tuning and performs reliably across a reasonable range

*Table 3.* Ablation study of PH-VAE on Pareto data. We vary the number of phases $m$ and the KL weight $\beta$. The default configuration used in all main experiments is $m = 10$, $\beta = 1$.

| Variant | $m$ | $\beta$ | $\text{KS}_{\text{tail}}$ ↓ | Q99 error ↓ |
| --- | --- | --- | --- | --- |
| Default (PH-VAE) | 10 | 1.0 | $0.084 \pm 0.006$ | $\mathbf{0.030 \pm 0.036}$ |
| PH-VAE (small) | 5 | 1.0 | $0.093 \pm 0.027$ | $0.217 \pm 0.177$ |
| PH-VAE (medium) | 15 | 1.0 | $\mathbf{0.055 \pm 0.008}$ | $0.149 \pm 0.254$ |
| PH-VAE ($\beta = 0.5$) | 10 | 0.5 | $0.110 \pm 0.008$ | $0.859 \pm 0.642$ |
| PH-VAE ($\beta = 2.0$) | 10 | 2.0 | $0.083 \pm 0.008$ | $0.078 \pm 0.067$ |

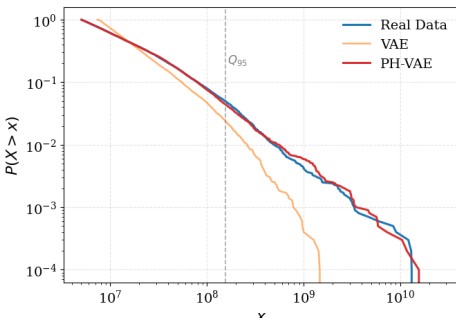

*Figure 5.* Log–log CCDF of word-frequency counts from the Google Web Trillion Word Corpus, comparing real data, Gaussian VAE, and PH-VAE.

of configurations. Thanks to the closed-form PH likelihood, PH-VAE trains with a cost comparable to standard VAEs and remains stable across a small range of $m$ and $\beta$.

Overall, these real-world univariate experiments illustrate that PH-VAE robustly captures heavy-tailed behavior across diverse domains within the empirically observed range of the data, where classical VAEs fail, motivating its extension to multivariate settings with complex dependence structures.

### 4.3. Multivariate Heavy-Tailed Modeling

We experiment with PH-VAE in multivariate settings, where the goal is not only to model heavy-tailed marginals, but also to capture cross-dimensional dependence and joint extreme behavior. Although the decoder factorizes conditionally on the latent variable (see Equation (4)), statistical dependence between dimensions is induced via the shared latent representation. This design enables PH-VAE to learn multivariate, correlated data without explicitly specifying a parametric copula or correlation structure, while still preserving independence when supported by the data.
PH-VAE does not aim to model asymptotic tail dependence or extreme-value limits. Approaches explicitly designed for that purpose (e.g., copula-based or EVT-based models) make fundamentally different assumptions and often sacrifice tractable likelihoods or end-to-end training. Our focus is instead on flexible, data-adaptive likelihoods within the VAE framework. We evaluate the proposed approach on both controlled synthetic multivariate datasets, where the ground-truth dependence structure is known, and real financial returns, which provides a challenging real-world benchmark with unknown and noisy dependence.

#### 4.3.1. SYNTHETIC MULTIVARIATE DATA

We first consider synthetic multivariate datasets with known heavy-tailed marginals and controlled dependence structure. Following copula-based synthetic data generation methods (Houssou et al., 2022), we generate data using Student-t copulas with heterogeneous marginals (Pareto, Weibull,

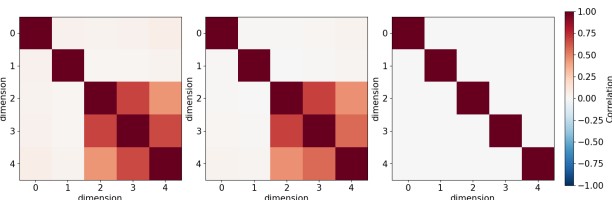

*Figure 6.* Illustration of correlation matrices of synthetic data, comparing ground truth (left), PH-VAE (middle), and Independent PH-VAEs (right).

Lognormal, and Burr), allowing us to disentangle marginal tail accuracy from dependence modeling. This setting enables quantitative evaluation of whether PH-VAE can simultaneously recover marginal tail behavior and multivariate dependence.

To isolate the mechanism responsible for dependence modeling, we compare PH-VAE against an *Independent PH-VAEs* baseline, in which each dimension is modeled by a separate univariate PH-VAE without a shared latent representation. Both models use identical architectures (same latent dimensionality and number of phases per dimension), ensuring that any performance differences arise solely from the presence or absence of latent sharing.

Across all dimensions, marginal tail accuracy in the multivariate setting remains comparable to the univariate case ($KS_{tail} \approx 0.02 - 0.04$; Q99 relative error below 15%), indicating that the multivariate setting does not degrade tail fidelity.

**Dependence structure and negative controls.** The synthetic setup is explicitly designed to include both dependent and independent dimension pairs. Dimensions 0 and 1 are generated as independent by construction, serving as negative controls to test whether the model introduces spurious dependence.

Figure 6 illustrates correlation matrices computed on $\log(1 + x)$-transformed data, following standard practice in heavy-tailed analysis to reduce the impact of outliers on correlation estimates (Cont, 2001). PH-VAE accurately recovers the ground-truth dependence structure: independent dimension pairs remain near zero correlation, while dependent pairs exhibit the correct magnitude and sign. Crucially, this behavior on the negative controls demonstrates that the shared latent representation does not impose artificial coupling, but instead induces dependence selectively where supported by the data. As expected, the Independent PH-VAEs baseline has independent dimensions, as illustrated also in Figure 6.

Table 4 reports correlation error, Kendall's $\tau$ error, and tail co-exceedance error for Gaussian VAE, Independent PH-VAEs, and multivariate PH-VAE. While correlation error is similar between Gaussian VAE and PH-VAE, PH-VAE substantially outperforms on Kendall's $\tau$ error and tail co-

*Table 4.* Synthetic multivariate (5D) data dependence metrics.

| Model | Corr. error ↓ | Kendall $\tau$ error ↓ | Tail CoExErr@99 ↓ |
|---|---|---|---|
| Gaussian VAE | 0.5798 | 0.1429 | $1.38\times10^{-3}$ |
| Independent PH-VAEs | 1.8824 | 0.2679 | $2.04\times10^{-3}$ |
| PH-VAE | **0.5659** | **0.0672** | **$5.44\times10^{-4}$** |

exceedance error. This is because these metrics are more sensitive to joint tail behavior, which PH-VAE captures more accurately through its more flexible tail modeling.

These results confirm that the shared latent variable is the primary mechanism through which PH-VAE models realistic dependence and joint extremes, without relying on an explicit copula specification.

### 4.3.2. REAL FINANCIAL RETURNS

We further evaluate PH-VAE on real multivariate financial data, consisting of absolute daily returns of five liquid U.S. equities (AAPL, MSFT, AMZN, GOOGL, and META) from January 2010 to December 2024 at daily (business-day) frequency. Financial returns are well known to exhibit heavy tails and nontrivial dependence, particularly during periods of market stress. Unlike the synthetic setting, the true data-generating distribution and dependence structure are unknown, and the evaluation therefore focuses on cross-sectional dependence and joint extreme behavior rather than goodness-of-fit to a parametric ground truth.

As shown in Table 5, PH-VAE consistently outperforms Gaussian VAE across all dependence metrics, particularly for Kendall's $\tau$ error. These results show that PH-VAE captures complex dependence in heavy-tailed financial data without explicit copula modeling. Note also that for this dataset, the Kendall $\tau$ error is much lower for Independent PH-VAEs than for Gaussian VAE. This is because Kendall's $\tau$ measures rank correlation and does not fully capture linear dependence structure. Indeed, while the Independent PH-VAEs model achieves low Kendall $\tau$ error, it exhibits large correlation error.

*Table 5.* Real financial returns dependence metrics.

| Model | Corr. error ↓ | Kendall $\tau$ error ↓ | Tail CoExErr@99 ↓ |
|---|---|---|---|
| Gaussian VAE | 0.8848 | 0.2667 | $1.38\times10^{-3}$ |
| Independent PH-VAEs | 1.8645 | 0.0359 | $2.05\times10^{-3}$ |
| PH-VAE | **0.1716** | **0.0293** | **$9.79\times10^{-4}$** |

Overall, these results show that PH-VAE extends naturally to multivariate settings, capturing both heavy-tailed marginals and realistic cross-dimensional dependence while avoiding spurious correlations.

**Limitations** Despite its strong empirical performance, PH-VAE has several limitations. First, accurate tail modeling requires sufficient exposure to extreme observations during training, when tail events are heavily truncated or rare,

extrapolation quality degrades, particularly for very high quantiles. Second, tail estimation becomes less stable in low-data regimes, where rare-event statistics are inherently difficult to estimate reliably. Third, in high-dimensional settings, limited latent capacity may reduce the model's ability to simultaneously capture marginal tail behavior and cross-dimensional dependence. Finally, although PH distributions can closely approximate heavy-tailed behavior over finite, data-relevant ranges, they remain asymptotically light-tailed. Consequently, PH-VAE is not designed to recover exact asymptotic power-law behavior, but rather to provide flexible and tractable modeling of empirically observed heavy-tail phenomena.

## 5. Conclusion and Future Work

This paper introduced the Phase-Type VAE, a novel generative model specifically designed to address the limitations of the VAE framework when modeling heavy-tailed distributions. The core innovation lies in replacing the traditional fixed-output decoder with a flexible PH distribution decoder. PH-VAE can adaptively capture a wide range of observed tail behaviors, from exponential decay to power-law dynamics. The model leverages the closed-form density of PH distributions to enable efficient training via a tailored ELBO. Experiments conducted on univariate and multivariate synthetic benchmarks, together with real-world observed data, demonstrate that PH-VAE significantly outperforms standard VAEs, accurately reconstructing both the body and the crucial tail of complex distributions. Future work will focus on extending the framework to high-dimensional data, including images and other distributions supported beyond $\mathbb{R}_+$.

## Acknowledgements

This work has received funding from the European Union's Horizon research and innovation program Chips JU under Grant Agreement No. 101139769, DistriMuSe project (HORIZON-KDT-JU-2023-2-RIA). The JU receives support from the European Union's Horizon research and innovation program and the nations involved in the mentioned projects. The work reflects only the authors' views; the European Commission is not responsible for any use that may be made of the information it contains.

## Impact Statement

This work introduces a new interface between deep generative modeling and applied probability by integrating Phase-Type distributions into variational autoencoders. The proposed PH-VAE demonstrates that structured stochastic-process likelihoods can be learned end-to-end within modern latent-variable models, enabling accurate modeling of heavy-tailed phenomena without committing to a fixed extreme-value family. Beyond improved tail fidelity, this perspective reframes decoder design as the learning of a generative mechanism rather than the selection of a parametric distribution. We expect this approach to stimulate further interaction between representation learning and classical probabilistic modeling, including extensions to structured time-to-event data, multivariate extremes, and other settings where rare events dominate system behavior. As with other general-purpose generative models, responsible deployment in high-stakes domains should consider appropriate validation and domain-specific safeguards, but we do not anticipate direct negative societal impacts arising from this work.

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

# A. Tail Evaluation Metrics

Accurately evaluating heavy-tailed generative models requires metrics that focus explicitly on the extreme-value region of the distribution. Classical global goodness-of-fit measures, such as the unconditional Kolmogorov-Smirnov (KS) distance, are dominated by discrepancies in the body of the distribution and are therefore poorly suited for assessing tail fidelity. We instead employ two complementary tail-focused metrics: a conditional tail KS distance and an extreme quantile error.

## A.1. Conditional Tail Kolmogorov-Smirnov Distance

Let $F_{\text{true}}$ denote the ground-truth cumulative distribution function (CDF) and $F_{\text{model}}$ the CDF induced by samples generated from a trained model. Let $q \in (0, 1)$ be a high quantile level, fixed to $q = 0.95$ in all experiments, and define the corresponding tail threshold

$$x_q = F_{\text{true}}^{-1}(q).$$

We define the conditional tail CDF associated with a distribution $F$ as

$$F(x \mid X \geq x_q) = \frac{F(x) - F(x_q)}{1 - F(x_q)}, \qquad x \geq x_q.$$

For the ground-truth distribution, $F_{\text{true}}(x_q) = q$ by definition, while for the model distribution, $F_{\text{model}}(x_q)$ is estimated empirically from generated samples.

The conditional tail Kolmogorov-Smirnov distance is then given by

$$\text{KS}_{\text{tail}} = \sup_{x \geq x_q} |F_{\text{model}}(x|X \geq x_q) - F_{\text{true}}(x|X \geq x_q)|.$$

In practice, $F_{\text{model}}(x|X \geq x_q)$ is computed empirically using samples generated from the model, restricted to values exceeding $x_q$ and renormalized to form a conditional empirical CDF. For synthetic datasets, $F_{\text{true}}$ is known analytically, allowing exact computation of the conditional tail CDF.

This metric isolates tail *shape* discrepancies and is invariant to errors in the body of the distribution, making it particularly well suited for heavy-tailed settings. For some model distribution pairs, extreme quantiles may be undefined due to tail collapse or insufficient tail mass; such cases are reported as N/A in Table 1.

## A.2. Extreme Quantile Error

While $\text{KS}_{\text{tail}}$ captures discrepancies in tail *shape*, it does not directly assess the accuracy of extreme-value *magnitudes*. We therefore additionally evaluate the accuracy of high-quantile estimation.

Let $Q_{0.99}^{\text{true}}$ and $Q_{0.99}^{\text{model}}$ denote the 99th percentiles of the true and model distributions, respectively. We consider the *relative* extreme quantile error as:

$$\text{Q99}_{\text{rel}} = \frac{|Q_{0.99}^{\text{model}} - Q_{0.99}^{\text{true}}|}{Q_{0.99}^{\text{true}}}.$$

In the experimental results, we report relative quantile errors, as they provide a scale-normalized measure that enables comparison across datasets with different magnitudes. We note that for heavy-tailed distributions, relative errors can take large values due to the growth of extreme quantiles. This behavior reflects genuine difficulties in extreme-value estimation rather than numerical instability.

## A.3. Estimation Protocol

For each trained model, tail metrics are estimated using 100,000 samples drawn from the generative model. All reported results are averaged over multiple independent training runs with different random seeds, and we report mean values together with one standard deviation.

## A.4. Limitations of KS-Based Metrics for Extremely Heavy Tails

For distributions with extremely slow decay, such as Zipf-like word-frequency distributions, the KS distance, even when restricted to the tail, may approach one for all models due to small discrepancies in support or discreteness effects. In such

cases, quantitative KS-based metrics become uninformative. We therefore complement numerical evaluation with qualitative diagnostics, including log-log complementary CDF plots, which provide a more reliable assessment of tail behavior in these regimes.

## B. Multivariate Dependence Metrics

This appendix provides formal definitions and estimation protocols for the multivariate dependence metrics used in Section 4.3. These metrics evaluate whether a generative model captures cross-dimensional dependence and joint extreme behavior beyond marginal tail fidelity.

### B.1. Correlation Error on Log-Transformed Data

Heavy-tailed data often exhibit infinite or highly unstable second moments, making raw-scale correlation estimates unreliable. Following standard practice in heavy-tailed and financial data analysis (Cont, 2001; Resnick, 2007), we therefore compute correlations on log-transformed data.

Given multivariate observations $x \in \mathbb{R}_+^D$, we apply the elementwise transformation

$$y = \log(1 + x),$$

and compute the empirical Pearson correlation matrix $C \in \mathbb{R}^{D \times D}$ from samples of $y$.

Let $C_{\text{real}}$ denote the correlation matrix estimated from real data, and $C_{\text{gen}}$ the corresponding matrix estimated from model-generated samples. The correlation error is defined as the Frobenius norm:

$$\text{CorrErr} = \|C_{\text{gen}} - C_{\text{real}}\|_F.$$

This metric captures discrepancies in linear dependence structure while mitigating the influence of extreme marginal values.

### B.2. Kendall's $\tau$ Error

To assess rank-based dependence, which is invariant to monotonic marginal transformations and robust to heavy-tailed marginals, we compute pairwise Kendall's $\tau$ coefficients.

For each pair of dimensions $(i, j)$, let $\tau_{ij}^{\text{real}}$ and $\tau_{ij}^{\text{gen}}$ denote the empirical Kendall's $\tau$ computed from real and generated samples, respectively. The Kendall error is defined as the average absolute deviation:

$$\text{TauErr} = \frac{2}{D(D-1)} \sum_{1 \leq i < j \leq D} \left| \tau_{ij}^{\text{gen}} - \tau_{ij}^{\text{real}} \right|.$$

This metric evaluates whether the model reproduces the rank-dependent structure of the data, independent of the marginal scale.

### B.3. Tail Co-Exceedance Probability

To explicitly probe joint extreme behavior, we measure tail co-exceedance probabilities. For a dimension pair $(i, j)$, the empirical tail co-exceedance probability at quantile level $q \in (0, 1)$ is defined as

$$\widehat{p}_{i,j}(q) = \frac{1}{N} \sum_{n=1}^{N} \mathbf{1}\left[ x_i^{(n)} > Q_i(q) \ \wedge \ x_j^{(n)} > Q_j(q) \right], \tag{10}$$

where $N$ denotes the number of samples and $Q_i(q)$ is the $q$-th marginal quantile of dimension $i$ estimated from real data.

Tail co-exceedance error is computed as the average absolute difference between real and generated co-exceedance probabilities across all unordered dimension pairs, see Equation (9).

This statistic isolates simultaneous extreme events that are not detectable through correlation or rank-based measures alone.

## B.4. Estimation Protocol

For all multivariate experiments, dependence metrics are computed using large batches of samples generated from the trained model (typically $10^6$ samples), ensuring stable estimation in the tail. Empirical quantiles and correlation statistics are computed independently for real and generated data. Reported values correspond to averages over multiple random seeds where applicable.

Together, these complementary metrics assess linear dependence, rank dependence, and joint tail behavior. providing a comprehensive evaluation of multivariate heavy-tailed modeling performance.

# C. Phase-Type Distributions and Heavy-Tail Approximation

### C.1. Formal Definition of Phase-Type Distributions

A *Phase-Type* distribution is defined as the probability distribution of the time to absorption in a continuous-time Markov chain (CTMC) with one absorbing state and a finite number of transient states. PH distributions were introduced by Neuts (Neuts, 1981) and have since become a central modeling tool in applied probability and performance evaluation.

Formally, consider a CTMC with $m$ transient states and one absorbing state. Let $\boldsymbol{\alpha} \in \mathbb{R}^m$ denotes the initial probability vector over the transient states, satisfying $\boldsymbol{\alpha} \geq 0$ and $\boldsymbol{\alpha}\mathbf{1} = 1$. Let $\mathbf{A} \in \mathbb{R}^{m \times m}$ denote the sub-generator matrix associated with the transient states, such that (i) $A_{ii} < 0$ for all $i$, (ii) $A_{ij} \geq 0$ for $i \neq j$, and (iii) $\mathbf{A}\mathbf{1} \leq \mathbf{0}$. The exit-rate vector (that is, the vector containing the transition rates from every transient state to the absorbing state) is obtained as $\mathbf{t} = -\mathbf{A}\mathbf{1}$.

The absorption time

$$X = \inf\{t > 0 : \text{the CTMC enters the absorbing state}\}$$

is said to follow a Phase-Type distribution with representation $(\boldsymbol{\alpha}, \mathbf{A})$, denoted $X \sim \mathrm{PH}(\boldsymbol{\alpha}, \mathbf{A})$.

Intuitively, a PH distribution models a random duration as the cumulative time spent traversing a sequence of latent *phases*, where each phase has an exponentially distributed holding time and transitions probabilistically to other phases before eventual absorption. This construction allows PH distributions to represent highly flexible, multi-modal, and skewed behaviors while retaining analytical tractability.

The pair $(\boldsymbol{\alpha}, \mathbf{A})$ fully characterizes the distribution of $X$. Different representations may correspond to the same distribution, a property known as non-identifiability, which is discussed further in C.4. Despite this, PH distributions form a mathematically well-defined and expressive family that admits closed-form expressions for key probabilistic quantities, making them particularly suitable for integration into likelihood-based generative models.

### C.2. Stochastic Process Associated with a PH Distribution

We consider a PH distribution with $m$ phases. The associated CTMC has $m$ transient and one absorbing state and the corresponding stochastic process evolves as follows. The system is initialized in one of the transient states according to the probability vector $\boldsymbol{\alpha}$. While in a transient state $i$, the process remains in that state for a random holding time $\tau_i$ that is exponentially distributed with rate $-A_{ii}$. Upon leaving state $i$, the process jumps to another transient state $j \neq i$ with probability

$$\mathbb{P}(i \to j) = \frac{A_{ij}}{-A_{ii}},$$

or is absorbed with probability

$$\mathbb{P}(i \to \mathrm{abs}) = \frac{t_i}{-A_{ii}},$$

where $t_i = -(\mathbf{A}\mathbf{1})_i$ denotes the exit rate from state $i$.

The total time to absorption is obtained by summing the holding times spent in each visited transient state along the realized path of the Markov chain. This accumulated duration defines a Phase-Type distributed random variable.

A sample from a given PH distribution can be obtained by simulating the stochastic process described above, as outlined in Algorithm 2. For the multivariate PH-VAE architecture, samples from multiple independent PH distributions are needed. This can be achieved by repeated application of Algorithm 2, as shown in Algorithm 3.

---

**Algorithm 2** Sampling from a Single Phase-Type Distribution

---

**Input:** PH parameters $(\alpha, A, t)$
**Output:** Sample $x \sim \text{PH}(\alpha, A)$
Sample initial phase $s \sim \text{Categorical}(\alpha)$  Set $x \leftarrow 0$
**while** *s is not absorbing* **do**
  Sample holding time $\Delta t \sim \text{Exp}(-A_{ss})$
  $x \leftarrow x + \Delta t$
  Sample next state according to the probabilities

$$\begin{cases} s \leftarrow j & \text{with prob. } A_{sj}/(-A_{ss}), \quad j = 1, 2, ..., m \\ s \leftarrow absorbing & \text{with prob. } t_s/(-A_{ss}) \end{cases}$$

**return** $x$

---

**Algorithm 3** Sampling from a Multivariate PH-VAE

---

**Input:** PH parameters $\{(\alpha_j, A_j, t_j)\}_{j=1}^{k}$
**Output:** Sample $x \in \mathbb{R}_+^k$
**for** $j = 1$ **to** $k$ **do**
  Sample $x_j \sim \text{PH}(\alpha_j, A_j, t_j)$ using Algorithm 2
**return** $x = (x_1, \ldots, x_k)$

---

This construction highlights the compositional nature of PH distributions: the absorption time can be interpreted as a random sum of exponential phases, where both the number of phases visited and their transition order are random. By appropriately configuring the transition structure and rates of the CTMC, PH distributions can capture a wide variety of distributional shapes, including strong skewness, multi-modality, and slowly decaying tails over finite ranges.

From a modeling perspective, the CTMC interpretation provides two key advantages. First, it endows PH distributions with a clear generative mechanism, making sampling straightforward via simulation of the underlying Markov chain, opening the door to explainability. Second, it yields closed-form expressions for likelihoods, cumulative distribution functions, and tail probabilities through matrix-exponential formulas, which are directly exploited in the PH-VAE decoder.

Importantly, the CTMC viewpoint also clarifies the distinction between *structural* expressiveness and *asymptotic* tail behavior. While the chain is finite and thus guarantees eventual absorption, the diversity of paths and holding times enables PH distributions to approximate complex heavy-tailed behavior over practically relevant ranges, a point further examined in Appendix C.7.

## C.3. Density, Distribution, and Tail Functions of Phase-Type Distributions

One of the main advantages of PH distributions is that their probabilistic quantities admit closed-form expressions derived from the underlying continuous-time Markov chain representation. These expressions are central to both theoretical analysis and practical likelihood-based learning.

Let $X \sim \text{PH}(\boldsymbol{\alpha}, \mathbf{A})$ be a Phase-Type distributed random variable with initial distribution $\boldsymbol{\alpha}$ and transient sub-generator matrix $\mathbf{A}$. Calculate the exit-rate vector as $\mathbf{t} = -\mathbf{A1}$. The probability density function (PDF) of $X$ is given by

$$f_X(x) = \boldsymbol{\alpha} \exp(\mathbf{A}x) \mathbf{t}, \quad x > 0. \tag{11}$$

The cumulative distribution function and complementary cumulative distribution function take the form

$$F_X(x) = 1 - \boldsymbol{\alpha} \exp(\mathbf{A}x) \mathbf{1}, \tag{12}$$

$$\bar{F}_X(x) = \boldsymbol{\alpha} \exp(\mathbf{A}x) \mathbf{1}. \tag{13}$$

These matrix-exponential expressions follow directly from the transient-state probability evolution of the underlying CTMC. In particular, $\exp(\mathbf{A}x)$ is the transition probability matrix of the CTMC among the transient states over a time period of length $x$. Consequently, $\boldsymbol{\alpha} \exp(\mathbf{A}x)\mathbf{1}$ is the probability that the process is still in one of the transient states after $x$ time units, that is, the probability that $X$ is greater than $x$.

The CCDF representation in Eq. (13) is particularly important for heavy-tail modeling, as it provides direct access to tail probabilities without numerical integration. This form enables precise evaluation of tail-focused metrics such as conditional Kolmogorov-Smirnov distances and extreme quantiles, which are central to the empirical evaluation of PH-VAE.

Beyond distribution functions, PH distributions possess additional analytical properties that are beneficial for learning and inference. All moments of $X$ exist and can be computed in closed form as

$$\mathbb{E}\left[X^k\right] = k!\,\boldsymbol{\alpha}(-\mathbf{A})^{-k}\mathbf{1}, \quad k \in \mathbb{N},$$

and the Laplace transform admits the expression

$$\mathcal{L}_X(s) = \boldsymbol{\alpha}(s\mathbf{I} - \mathbf{A})^{-1}\mathbf{t}, \quad s > 0.$$

These closed-form expressions highlight a key distinction between Phase-Type distributions and many classical heavy-tailed families: although PH distributions are defined through finite-state Markov chains, they retain exact and numerically stable formulas for likelihoods and tail probabilities. This analytical tractability enables their seamless integration into the PH-VAE decoder, where the log-likelihood in Eq. (11) is evaluated directly during training without resorting to numerical approximations or discretization.

### C.4. Acyclic (Series Canonical) Phase-Type Representations and Parameter Efficiency

A given Phase-Type (PH) distribution admits an infinite number of distinct Markovian representations that induce the same absorption-time distribution (Horváth & Telek, 2024). As a consequence, a given PH distribution may be represented by generator matrices of different structure and dimensionality.

Moreover, the $(\boldsymbol{\alpha}, \mathbf{A})$ representation is over-parameterized. While it contains $m^2 + m - 1$ free parameters for an $m$-phase PH distribution ($m - 1$ parameters in the initial probability vector, which must sum to one, and $m^2$ in the generator matrix) the true number of degrees of freedom is only $2m - 1$. This follows from the fact that the first $2m - 1$ moments uniquely determine a PH distribution with $m$ phases (Horváth & Telek, 2024). Consequently, while the $(\boldsymbol{\alpha}, \mathbf{A})$ representation is mathematically valid and convenient for theoretical analysis, it is poorly suited for learning-based applications and optimization.

An acyclic PH distribution is a PH distribution whose transition graph contains no cycles. The generator matrix of an acyclic PH distribution can be brought into triangular form by a suitable reordering of the transient states (Cumani, 1982). Such a representation has $m - 1$ free parameters in the initial probability vector and $m(m + 1)/2$ in the generator; for a total of $(m^2 + 3m - 2)/2$ parameters.

A classical result is that any acyclic PH distribution admits a *series canonical representation*, also called the *series canonical form* (Cumani, 1982). In this representation, there are no restrictions on the initial probability vector $\boldsymbol{\alpha}$, but the generator $\mathbf{A}$ must follow the scheme

$$A_{ii} = -\lambda_i, \qquad A_{i,j} = \lambda_i \text{ if } j = i + 1, \qquad \text{and } A_{ij} = 0 \text{ otherwise,}$$

with ordered rates $0 < \lambda_1 \leq \lambda_2 \leq \cdots \leq \lambda_m$. The series canonical form is depicted in Figure 7. The following theorem states this result formally.

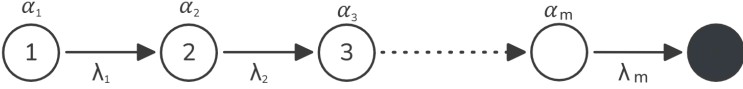

*Figure 7.* Series canonical form.

**Theorem. (Series Canonical Representation of Acyclic PH Distributions.)** Let $X$ be an acyclic Phase-Type distributed random variable with $m$ phases. Then there exists a Phase-Type distribution $\widetilde{X}$ with $m$ phases in *series canonical form* such that $X$ and $\widetilde{X}$ have the same distribution.

A proof and construction algorithm are given in (Cumani, 1982).

The series canonical structure yields substantial parameter savings: it has $2m - 1 \in O(m)$ parameters ($m - 1$ in the initial probability vector and $m$ in the generator) instead of the $(m^2 + 3m - 2)/2 \in O(m^2)$ parameters of a general acyclic representation. Moreover, the number of parameters in the series canonical form, $2m - 1$, equals the true number of degrees of freedom of a PH distribution.

From a learning and optimization perspective, this structural restriction is highly beneficial. Reducing the dimensionality of the decoder output lowers the risk of overfitting, simplifies constraint enforcement, and improves numerical stability. Moreover, using a canonical form removes the problem of non-identifiability inherent in general Phase-Type distributions. As discussed earlier, infinitely many different $(\boldsymbol{\alpha}, \mathbf{A})$ pairs can represent the same general PH distribution. This non-uniqueness is eliminated in the series canonical form, where the representation is unique up to degenerate cases in which some phases are never visited.

For these reasons, PH-VAE restricts the decoder to series canonical (acyclic) Phase-Type representations throughout all experiments. In the following section we show that this restriction does not limit expressive power, since acyclic PH distributions form a dense family on $(0, \infty)$.

### C.5. Universality of Phase-Type and Acyclic Phase-Type Distributions on $\mathbb{R}_+$

A fundamental theoretical result is that the class of Phase-Type (PH) distributions is *dense* (in the sense of weak convergence) in the set of all probability distributions on $(0, \infty)$. This implies that any positive-valued distribution can be approximated arbitrarily well by PH distributions. For a formal statement with proof see Theorem 4.2 on page 84 of (asm, 2003). The proof technique used there directly shows that the class of acyclic PH distributions is also dense on $(0, \infty)$.

Intuitively, this universality arises from the fact that PH distributions can be viewed as mixtures and convolutions of exponential distributions with flexible transition structures. By increasing the number of phases and appropriately configuring the generator matrix, PH distributions can adapt to approximate arbitrarily complex distributional shapes on $\mathbb{R}_+$, including strong skewness, multi-modality, and slowly decaying tails over finite ranges. A practical illustration is shown in Figure 8 where the PDF of a Weibull distribution is approximated by acyclic PH distributions with increasing number of phases.

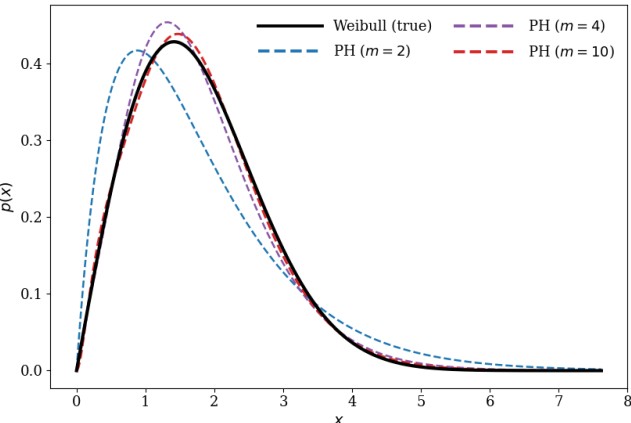

*Figure 8.* Practical illustration of approximating a PDF with Phase-Type distributions with different number of phases $m$.

The fact that the PH class is dense on $(0, \infty)$ does not imply that PH distributions inherit the asymptotic tail behavior of the target distribution. In particular, while PH distributions can closely approximate any tail behavior over any bounded interval, their asymptotic tail decay remains exponential due to the finite-state nature of the underlying Markov chain. This distinction is central to understanding the scope and limitations of PH-based modeling and is examined in detail in Appendix C.7.

In the context of generative modeling, universality ensures that PH-based decoders are not restricted to a predefined parametric family. Rather, the expressive power of the decoder increases systematically with the number of phases, enabling data-driven learning of distributional structure instead of manual specification of parametric forms.

## C.6. Approximation of Heavy-Tailed Distributions with Finite Phase-Type Models

Let $F$ denote a target heavy-tailed distribution on $\mathbb{R}_+$, such as a Pareto, Weibull (with shape parameter $< 1$), or Lognormal distribution. Although such distributions exhibit sub-exponential tail decay, finite-state PH distributions can approximate their cumulative distribution functions and tail probabilities with high accuracy over any bounded interval $[0, x_{\max}]$ by suitably increasing the number of phases and adjusting transition rates (Feldmann & Whitt, 1998; Horváth & Telek, 2000).

From a constructive perspective, this approximation arises from the ability of PH distributions to combine multiple exponential time scales through their underlying Markov chain structure. Paths that traverse many transient states with small exit rates generate rare but large absorption times, mimicking the behavior of heavy-tailed random variables over finite ranges. As the number of phases increases, the effective tail region over which the PH approximation remains accurate expands accordingly. Figure 9 shows an example of approximating a Burr CCDF.

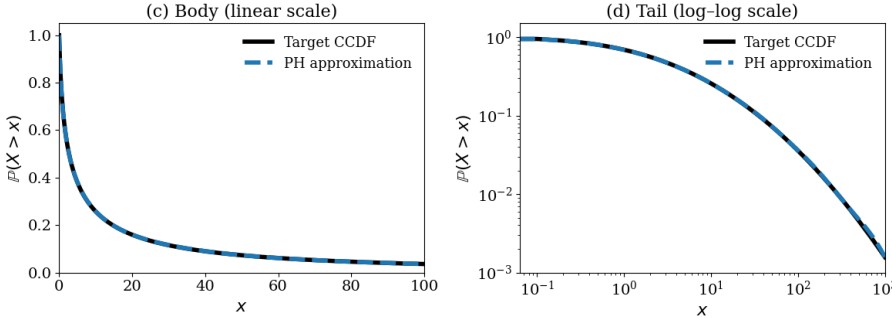

*Figure 9.* Approximation of a heavy-tailed target distribution, a Burr in this scenario, using finite Phase-Type models.

Importantly, the quality of the approximation should be understood in terms of finite-sample and finite-quantile accuracy rather than asymptotic equivalence. In practical learning scenarios, models are evaluated on observed data with bounded support and through metrics that emphasize extreme but finite quantiles, such as tail Kolmogorov-Smirnov distances and high-quantile errors. Finite PH models can achieve low discrepancy under such metrics even when the target distribution is heavy-tailed in the strict asymptotic sense.

This perspective aligns naturally with the PH-VAE framework. The decoder does not aim to recover an exact parametric form of the data-generating distribution but instead learns a finite PH representation that best matches the empirical distribution over the observed range. Increasing the number of phases increases expressiveness in a controlled manner, allowing the model to trade off complexity and approximation accuracy, as confirmed empirically by the ablation studies in Section 4.2.

In summary, finite Phase-Type distributions provide a flexible and tractable mechanism for approximating heavy-tailed behavior in practice. Their effectiveness stems not from reproducing the asymptotic tail class of the target distribution, but from accurately matching distributional shape and tail probabilities over the finite regimes encountered in real data. The implications of this distinction are examined more formally in Appendix C.7.

## C.7. Asymptotic vs. Practical Tail Behavior: Clarifying the Scope of Approximation

From a theoretical standpoint, all Phase-Type distributions are *light-tailed* in the strict asymptotic sense. Since the underlying continuous-time Markov chain has a finite number of states, the tail of any PH distribution ultimately decays exponentially. Formally, there exists a constant $\lambda > 0$ such that

$$\lim_{x \to \infty} e^{\lambda x} \, \bar{F}_X(x) = c \in (0, \infty).$$

Consequently, PH distributions do not belong to classical heavy-tailed families such as Pareto or regularly varying distributions when tail behavior is defined asymptotically as $x \to \infty$.

However, asymptotic tail classification is often misaligned with the objectives of data-driven modeling. In practical learning scenarios, models are trained and evaluated on finite datasets with bounded support, and performance is assessed through metrics that focus on extreme but finite quantiles, such as tail-based distances or high-percentile errors. In this setting, what matters is not the limiting behavior of the distribution at infinity, but rather its ability to accurately represent tail probabilities over the *observed and relevant range* of the data.

Finite Phase-Type distributions are particularly effective in this regime. Through the combination of multiple exponential phases and diverse transition paths in the underlying Markov chain, PH distributions can exhibit slow, heavy-tail-like decay over wide finite intervals. Rare paths involving many phases with small exit rates produce large absorption times with non-negligible probability, closely mimicking the empirical behavior of heavy-tailed distributions over several orders of magnitude.

This distinction resolves the apparent contradiction between the finite-state nature of PH distributions and their empirical success in modeling heavy-tailed data. The goal of PH-based modeling is not to reproduce the exact asymptotic tail class of the underlying data-generating process, which is typically unobservable, but to accurately approximate tail probabilities and extreme quantiles within the finite regime relevant for inference and prediction.

For the PH-VAE framework, this perspective is particularly natural. The decoder learns a finite PH representation that best matches the empirical distribution induced by the data and the latent space. As demonstrated in the experiments, such representations are sufficient to recover tail shape, tail mass, and extreme quantiles with high accuracy, despite the asymptotic light-tailed nature of PH distributions. This separation between asymptotic theory and practical modeling scope underlies the effectiveness and robustness of the proposed approach.

### C.8. Implications for Learning Heavy-Tailed Distributions in Variational Autoencoders

The theoretical properties of PH distributions discussed in this appendix have direct and important implications for learning heavy-tailed distributions within the VAE framework. In particular, PH distributions provide a principled mechanism for decoupling latent representation learning from tail-shape specification.

In standard VAEs, the decoder is typically restricted to a fixed parametric family, such as a Gaussian or Student-$t$ distribution. While such choices simplify optimization, they impose a predefined tail behavior that cannot adapt flexibly to the data. As a result, the latent space must compensate for mismatches in the decoder distribution, often leading to distorted representations and systematic underestimation of extreme events.

By contrast, a PH-based decoder shifts the modeling burden from selecting a specific heavy-tailed family to learning a structured generative mechanism. Conditioning the PH parameters on the latent variable allows the decoder to adapt the effective distributional shape, including tail mass and curvature, in a data-driven manner. The latent space thus captures meaningful variability in the data, while the PH decoder accounts for skewness and extreme behavior through its internal phase structure.

Importantly, the use of finite PH representations aligns naturally with the objectives of variational learning. VAEs are optimized using finite samples and evaluated through likelihoods defined over bounded ranges. As shown in Appendix C.7, PH distributions are particularly well suited to this setting, as they can closely approximate heavy-tailed behavior over the observed data range while retaining analytical tractability and stable optimization.

From an optimization perspective, the closed-form likelihoods of PH distributions enable exact evaluation of the reconstruction term in the ELBO, avoiding sampling-based approximations or discretization of the observation space. Combined with structured parameterizations such as acyclic representations, this results in stable training dynamics and scalable learning, as detailed in Appendix D.

In summary, Phase-Type distributions offer a theoretically grounded and practically effective decoder family for variational autoencoders operating on positive, skewed, and heavy-tailed data. By leveraging their universality, finite-range approximation power, and analytical tractability, PH-VAE provides a flexible alternative to fixed-family decoders, enabling accurate modeling of extreme events without sacrificing stability or interpretability.

## D. Training Stability and Computational Complexity

This appendix details the numerical stability mechanisms, optimization strategy, and computational complexity of the proposed PH-VAE model. These considerations are essential when integrating matrix-exponential likelihoods into deep generative models.

## D.1. Decoder Parameterization and Constraint Enforcement

The PH-VAE decoder outputs the parameters of a Phase-Type (PH) distribution conditioned on a latent variable $z$. In all experiments, an acyclic PH representation in series canonical form with $m$ phases is employed.

The decoder produces:

- an initial probability vector $\boldsymbol{\alpha}(z) \in \mathbb{R}^m$, enforced via a softmax transformation;

- a vector of positive transition rates $\lambda(z) = (\lambda_1, \ldots, \lambda_m)$.

Positivity and ordering constraints are enforced deterministically. Raw decoder outputs are transformed using a softplus activation followed by a cumulative sum. This guarantees $0 < \lambda_1 \leq \cdots \leq \lambda_m$ and prevents degenerate or ill-conditioned generator matrices.

The PH sub-generator matrix $\mathbf{A} \in \mathbb{R}^{m \times m}$ is assembled as an upper bidiagonal matrix:

$$A_{ii} = -\lambda_i, \qquad A_{i,j} = \lambda_i \text{ if } j = i + 1, \qquad \text{and } A_{ij} = 0 \text{ otherwise.}$$

The exit-rate vector is defined as $t = -A\mathbf{1}$, which in our case reduces to $t_m = \lambda_m$ and zero elsewhere. This construction ensures that $\mathbf{A}$ is a valid transient generator matrix for all decoder outputs.

## D.2. Numerical Stability of Latent Sampling

The encoder follows a standard Gaussian variational formulation. To prevent numerical instabilities during training, the log-variance output $\log \sigma^2$ is explicitly clamped to a bounded interval:

$$\log \sigma^2 \in [-30, \ 20].$$

This avoids overflow in the exponential operation used to compute the standard deviation and prevents excessively large latent perturbations during reparameterization. Empirically, this stabilization was sufficient to prevent NaNs or divergence across all experiments.

## D.3. PH Likelihood Evaluation

Phase-Type likelihoods involve matrix exponentials of the sub-generator matrix $\mathbf{A}$. Direct evaluation of $\exp(Ax)$ can be numerically unstable for large $x$ or during gradient-based optimization. To address this, we evaluate all matrix exponentials using *uniformization* (also known as randomization) (Stewart, 1994), a classical technique for continuous-time Markov chains.

Given a sub-generator matrix $\mathbf{A}$, we choose a uniformization rate

$$\Lambda \ \geq \ \max_i(-A_{ii}), \tag{14}$$

and define the corresponding discrete-time transition matrix

$$P \ = \ I + \frac{\mathbf{A}}{\Lambda}. \tag{15}$$

The matrix exponential admits the exact representation

$$\exp(Ax) = \sum_{k=0}^{\infty} e^{-\Lambda x} \frac{(\Lambda x)^k}{k!} P^k, \tag{16}$$

which expresses the continuous-time evolution as a Poisson-weighted sum of matrix powers. In practice, the series in (16) is truncated adaptively at the smallest $K$ such that the remaining Poisson tail mass is below a tolerance $\epsilon = 10^{-8}$. For a PH distribution with initial distribution $\alpha$ and exit rates $t = -A\mathbf{1}$, the probability density function is

$$f(x) = \alpha \exp(Ax)t, \tag{17}$$

which we evaluate by substituting the uniformized form of $\exp(Ax)$. This yields an exact and fully differentiable likelihood evaluation.

*Table 6.* Wall-clock training time and final Negative Log Likelihood (NLL) of PH-VAE as a function of the number of phases $m$ across heavy-tailed distributions. Training time per epoch remains essentially constant as $m$ increases, indicating that uniformization enables stable likelihood evaluation without introducing a noticeable computational overhead.

| Data | $m$ | Time / epoch (s) | Total time (s) | Final NLL $\downarrow$ |
|---|---|---|---|---|
| Weibull ($k = 0.8$) | 2 | $5.40 \pm 0.25$ | 72.5 | 1.0895 |
| | 4 | $5.18 \pm 0.16$ | 67.6 | 1.0863 |
| | 8 | $5.29 \pm 0.24$ | 69.1 | 1.0866 |
| | 16 | $5.23 \pm 0.17$ | 68.1 | **1.0863** |
| | 32 | $5.20 \pm 0.11$ | 67.3 | 1.0877 |
| Lognormal ($\sigma = 1.5$) | 2 | $5.23 \pm 0.19$ | 68.5 | **1.8255** |
| | 4 | $5.28 \pm 0.15$ | 68.3 | 1.8897 |
| | 8 | $5.16 \pm 0.12$ | 67.0 | 1.8846 |
| | 16 | $5.18 \pm 0.18$ | 67.5 | 1.8821 |
| | 32 | $5.13 \pm 0.07$ | 67.1 | 1.8822 |
| Burr ($c=1.5, k=0.8$) | 2 | $5.22 \pm 0.14$ | 68.0 | 1.5059 |
| | 4 | $5.32 \pm 0.21$ | 68.6 | 1.4988 |
| | 8 | $5.24 \pm 0.22$ | 67.8 | 1.4556 |
| | 16 | $5.25 \pm 0.18$ | 67.7 | 1.4552 |
| | 32 | $5.51 \pm 0.33$ | 71.3 | **1.4517** |
| Pareto ($\alpha = 2.4$) | 2 | $5.12 \pm 0.04$ | 67.1 | 1.2670 |
| | 4 | $5.19 \pm 0.12$ | 67.5 | 1.1111 |
| | 8 | $5.27 \pm 0.26$ | 69.4 | 0.9852 |
| | 16 | $5.29 \pm 0.23$ | 68.5 | 0.7973 |
| | 32 | $5.20 \pm 0.12$ | 67.3 | **0.6182** |

## D.4. Optimization and Training Stability

PH-VAE is trained using the Adam optimizer with a fixed initial learning rate and weight decay. To ensure stable optimization, gradient norms are clipped to a maximum value at each update step. This prevents rare large gradients, potentially induced by extreme observations from destabilizing training. A manual learning-rate decay schedule is employed, reducing the learning rate by a factor of $0.1$ every ten epochs.

## D.5. Discussion

These results demonstrate that incorporating a PH decoder into the VAE framework does not compromise numerical stability or scalability. Across all experiments, PH-VAE training was stable and reproducible across random seeds, without requiring specialized numerical solvers, second-order optimization methods, or problem-specific regularization techniques beyond standard deep-learning practices.

Table 6 provides a detailed empirical analysis of the impact of the number of PH phases $m$ on both optimization efficiency and model fit. Notably, the wall-clock training time per epoch remains essentially constant as $m$ increases across all data distributions. This behavior confirms that uniformization enables stable and efficient likelihood evaluation, preventing the theoretical increase in computational complexity from translating into practical runtime overhead.

