# OpenReview forum: "Phase-Type Variational Autoencoders for Heavy-Tailed Data"
_ICML.cc/2026/Conference — ICML 2026 regular_

### Official Review · Reviewer_RVRV · 2026-03-11

**Soundness:** 3
**Presentation:** 2
**Significance:** 3
**Originality:** 2
**Overall Recommendation:** 4
**Confidence:** 4

**Summary:**

This paper proposes PH-VAE, a VAE whose decoder likelihood is modeled with Phase-Type distributions instead of Gaussian ones, aiming to address the limited ability of standard VAEs to model heavy-tailed data. Experiments on synthetic heavy-tailed distributions and real datasets demonstrate improved tail modeling performance compared with Gaussian and other heavy-tail-aware VAE baselines.

**Compliance With Llm Reviewing Policy:**

Affirmed.

**Final Justification:**

The authors have addressed my major concerns during the rebuttal, so I decide to raise my score.

**Key Questions For Authors:**

1. Phase-Type distributions are asymptotically light-tailed and only approximate heavy-tailed behavior over a finite range. Could the authors clarify how PH-VAE models truly heavy-tailed phenomena, and under what conditions this approximation remains valid for extreme-event modeling?
2. The experiments mainly compare PH-VAE with Gaussian VAE and a few heavy-tail-aware variants. It would be helpful to include comparisons with more expressive likelihood models or generative models, or clarify why such baselines are not considered.
3. Most experiments are conducted on relatively low-dimensional datasets. How does PH-VAE scale to higher-dimensional or more complex data (e.g., larger tabular datasets or other modalities)?

**Limitations:**

Yes.

**Strengths And Weaknesses:**

Strengths:
- The paper addresses the problem of modeling heavy-tailed data within the VAE framework, which stems from a known limitation of standard VAE decoders that typically assume Gaussian likelihoods.
- The empirical results on synthetic and real datasets suggest improvements in tail modeling compared with Gaussian and other heavy-tail-aware VAE baselines.

Weaknesses:
- The main methodological contribution lies in replacing the Gaussian decoder likelihood with a Phase-Type distribution, while the overall framework and training objective remain largely standard VAE formulations. As a result, the degree of methodological novelty appears somewhat limited.
- Phase-Type distributions are asymptotically light-tailed and only approximate heavy-tailed behavior over a finite range, which raises questions about the theoretical justification for modeling truly heavy-tailed phenomena.
- The experimental evaluation lacks stronger baselines (e.g., more expressive generative models or alternative likelihood parameterizations), especially in multi-variant scienarios, making it difficult to fully assess the advantages of the proposed approach.

---

> ### Author Rebuttal · Authors · 2026-03-30
>
> We thank the reviewer for their constructive feedback. We appreciate the recognition of the importance of heavy-tail modeling and the encouraging empirical results. Clarifications:
> ### 1. Methodological Novelty
> We respectfully clarify that the contribution goes beyond a simple replacement of the Gaussian decoder.
>
> PH-VAE introduces a **Markov Chain based Phase-Type decoder**, where:
> - the likelihood corresponds to the **absorption time of a continuous-time Markov chain**,
> - and training relies on exact likelihood computation via matrix-exponential structure optimzied via **uniformization**.
>
> This results in a constructive and **highly flexible** likelihood family, fundamentally different from fixed parametric choices such as Gaussian, Student-t, or EVT-based models, while remaining fully tractable within the VAE framework.
>
> Another key novelty lies in introducing a learnable likelihood class capable of **adapting to a wide range of distributional shapes**, including complex and heavy-tailed behaviors, without requiring manual specification of a parametric family.
>
> To the best of our knowledge, this is the first work to incorporate Phase-Type distributions as decoder likelihoods in deep generative models, **enabling a principled bridge between classical applied probability and modern representation learning.**
>
>
> ### 2. Heavy-Tail Modeling with PH Distributions
> We fully agree that PH distributions are **asymptotically light-tailed**.
>
> The family of acyclic PH distributions is dense in the set of all probability distributions on R_+ (in the sense of weak convergence). We cite in our paper the corresponding theorem on page 4 as "Thm. 4.2 on page 84 of (Asmussen, 2003)" where also the proof is given. This guarantees, from a theoretical point of view, that any distribution on $\mathbb{R}_+$ can be approximated arbitrarily well by an acyclic PH distribution.
>
> As pointed out by Reviewers kovG and RVRV, and stated also in our paper on page 3, PH distributions are asymptotically light-tailed (they have exponential tail decay). It was however shown in a number of papers that, by appropriate combination of exponential scales, acyclic PH distributions are able to capture different types of heavy tailed behaviors over any finite interval. The seminal paper in this direction is (Feldmann et al., 1998) where proposed technique is limited to distributions with monotone decreasing probability density function. In our paper we cited also (Horvath & Telek, 2000) where this limitation is removed. Both techniques allow us to set an upper limit up to which the PH density follows a given heavy-tailed behavior. An illustration is given in Appendix C.6. A further discussion about "Asymptotic vs. Practical Tail Behavior" is provided in Appendix C.7.
>
> Empirically, we observe that this results in:
> - significantly improved estimation of **extreme quantiles**,
> - and better **tail shape fidelity** (e.g., log-CCDF behavior),
>
> We will revise the wording in the paper to make this scope explicit and avoid overstating asymptotic heavy-tail modeling.
>
>
> ### 3. Baselines and Comparative Evaluation
>
> We fully agree that the initial set of baselines was not sufficiently broad.
>
> Following the reviewer’s suggestion, We have extended the evaluation to include additional positive-support likelihoods (Lognormal and Gamma decoders).
>
> | Dataset   | KS_tail (Gamma-VAE) | KS_tail (LogNormal-VAE) | KS_tail (PH-VAE) | Q99_err (Gamma-VAE) | Q99_err (LogNormal-VAE) | Q99_err (PH-VAE) |
> |-----------|--------------------|-------------------------|------------------|---------------------|--------------------------|------------------|
> | lognormal | 0.136 ± 0.048      | **0.026 ± 0.010**       | 0.039 ± 0.013    | 0.306 ± 0.050       | 0.082 ± 0.051            | **0.066 ± 0.071**|
> | pareto    | 0.236 ± 0.100      | 0.072 ± 0.046           | **0.090 ± 0.030**| 0.342 ± 0.030       | 0.197 ± 0.079            | **0.086 ± 0.032**|
> | weibull   | 0.056 ± 0.004      | 0.326 ± 0.023           | **0.019 ± 0.004**| 0.082 ± 0.012       | 2.088 ± 0.381            | **0.025 ± 0.016**|
>
> The resulst confirms that the observed gains are robust across stronger baselines.
>
> ### 4. Scalability and Higher-Dimensional Data
> The current evaluation focuses on:
> - **univariate data**, and
> - **moderate-dimensional multivariate settings** (synthetic copulas and real financial data),
>
> As suggested by the reviewer, we conducted additional experiments to assess the behavior of PH-VAE in higher-dimensional settings.
>
> | dim | latent | corr err | q90 err | q95 err | q99 err |
> |-----|--------|----------|--------|--------|--------|
> | 20  | 10     | 4.42     | 0.08   | 0.10   | 0.31   |
> | 50  | 10     | 8.67     | 0.07   | 0.13   | 0.34   |
> | 500 | 20     | 92.48    | 1.17   | 1.27   | 1.93   |
>
> Overall, these results confirm that PH-VAE scales well in the regimes targeted in this work, with clear directions for extending it to even higher-dimensional settings.

---

> > ### Author Rebuttal · Reviewer_RVRV · 2026-04-03
> >
> > Thank you for the detailed rebuttal. I appreciate the clarification of the PH decoder construction, which shows that the approach is not a trivial plug-in but requires careful design for tractable and stable likelihood computation. The revised positioning as finite-range heavy-tail modeling and the additional empirical results are also helpful. I acknowledge that the problem is important and the method is technically sound. Overall, my concerns are partially addressed, and I will raise my score.

---

> > > ### Author Response · Authors · 2026-04-05
> > >
> > > Thank you for your thoughtful feedback and for the positive assessment. We appreciate your time and consideration.

---

### Official Review · Reviewer_5ABB · 2026-03-13

**Soundness:** 4
**Presentation:** 4
**Significance:** 3
**Originality:** 4
**Overall Recommendation:** 5
**Confidence:** 4

**Summary:**

The paper introduces a novel constructive likelihood (eg: distribution family with closed-form sampling and density) for variational autoencoders, built out of the phase-type distribution for the absorption time of a Markov chain. After demonstrating the construction, the paper evaluates by comparison to Student's t-likelihood and extreme-value likelihood VAEs on metrics for accuracy of distribution approximation at the tails, on synthetic and real-world univariate data.

**Compliance With Llm Reviewing Policy:**

Affirmed.

**Key Questions For Authors:**

What do the numerical results look like when including 95% confidence intervals or standard errors?  Please note that my current score is pending clarification on this issue, and that if the results do not stand up when confidence intervals are added, I will lower my score.

**Limitations:**

I would prefer if the authors could provide a little more discussion of why they left applications to multidimensional data for future work.

**Strengths And Weaknesses:**

Strengths:
* Interesting construction using PH distributions.
* Nice find regarding the universal approximation property
* Reasonable evaluation metrics and performance for the new model
* Very encouraging to see that best results are obtained with $\beta=1$, indicating that the KL penalty performs as designed and does not require annealing during training.

Weaknesses:
* No multivariate/multidimensional datasets in evaluation.
* No confidence intervals are presented for numerical metrics, making claims of superior performance difficult to really quantify.

---

> ### Author Rebuttal · Authors · 2026-03-30
>
> We sincerely thank the reviewer for their thoughtful and encouraging feedback. We greatly appreciate the positive assessment of the construction, theoretical properties, and empirical results.
>
> We address all the remaining concerns below:
>
> ### 1. Confidence Intervals and Statistical Reliability
>
> We fully agree that reporting uncertainty estimates is important. Only some tables in the paper report standard deviation.
>
> In response, we are re-running all experiments with many independent seeds and will report **95% confidence intervals** for all numerical metrics.
> For synthetic data:
> | Model        | Metric              | Mean ± Std        | 95% CI              |
> |--------------|---------------------|-------------------|---------------------|
> | VAE          | corr_error          | 0.511 ± 0.099     | [0.414, 0.608]      |
> | VAE          | kendall_tau_error   | 0.104 ± 0.024     | [0.083, 0.125]      |
> | VAE          | tail_coex_abs_error | 0.002 ± 0.000     | [0.001, 0.002]      |
> | PH-VAE-MULTI | corr_error          | 0.175 ± 0.018     | [0.159, 0.192]      |
> | PH-VAE-MULTI | kendall_tau_error   | 0.026 ± 0.006     | [0.021, 0.031]      |
> | PH-VAE-MULTI | tail_coex_abs_error | 0.001 ± 0.000     | [0.000, 0.001]      |
>
> For real data:
>
> | Model         | Metric                | Mean ± Std       | 95% CI              |
> |---------------|-----------------------|------------------|---------------------|
> | VAE           | corr_error            | 0.709 ± 0.156    | [0.572, 0.846]      |
> | VAE           | kendall_tau_error     | 0.100 ± 0.027    | [0.077, 0.125]      |
> | VAE           | tail_coex_abs_error   | 0.002 ± 0.000    | [0.002, 0.002]      |
> | PH-VAE-MULTI  | corr_error            | 0.210 ± 0.071    | [0.148, 0.271]      |
> | PH-VAE-MULTI  | kendall_tau_error     | 0.033 ± 0.008    | [0.026, 0.040]      |
> | PH-VAE-MULTI  | tail_coex_abs_error   | 0.001 ± 0.000    | [0.001, 0.002]      |
>
> and for univarite, the paper reports mean ± standard deviation across seeds. For completeness, we now additionally report standard errors (σ/√n) below. The conclusions remain unchanged.
>
> | Model  | Weibull (KS/Q99) | Pareto (KS/Q99) | Lognormal (KS/Q99) | Burr (KS/Q99) |
> |--------|------------------|-----------------|--------------------|---------------|
> | VAE    | 0.190±0.007 / 0.235±0.002 | 0.259±0.003 / 0.336±0.001 | 0.498±0.004 / 0.582±0.001 | 0.433±0.001 / 0.724±0.002 |
> | t-VAE  | 0.225±0.052 / 0.878±0.000 | 0.233±0.014 / 0.632±0.003 | 0.650±0.001 / 100.783±0.971 | 0.518±0.001 / 39.845±1.061 |
> | xVAE   | 0.187±0.005 / 0.062±0.003 | 0.161±0.001 / 0.131±0.005 | 0.032±0.003 / 0.209±0.012 | N/A / 1.000±0.000 |
> | PH-VAE | **0.022±0.002 / 0.010±0.001** | **0.051±0.002 / 0.023±0.007** | **0.020±0.003 / 0.016±0.005** | **0.080±0.005 / 0.599±0.005** |
>
>
>
>
> ### 2. Multivariate / Multidimensional Scope
>
> We thank the reviewer for raising this important point.
>
> The current scope of the paper includes:
> - **synthetic multivariate datasets** (e.g., copula-based constructions),
> - **5D real-world financial data**,
>
> where the goal is to study **dependence and joint tail behavior** through the shared latent representation.
>
> We chose this setting deliberately to isolate and analyze tail modeling and dependence in a controlled way, where a specific dependence structure exists between dimensions 2, 3, and 4, while dimensions 0 and 1 remain independent (as illustrated by the confusion matrices in Fig. 6).
>
> The goal is to evaluate whether PH-VAE captures joint tail behavior through the shared latent representation. We agree that very high-dimensional settings remain challenging and explicitly list this as a limitation and direction for future work.
>
> We will clarify this scope and limitation explicitly in the final version.
>
> We again sincerely thank the reviewer for the constructive feedback. We hope that the addition of confidence intervals and the clarification of the multivariate scope fully address the remaining concerns.

---

> > ### Author Rebuttal · Reviewer_5ABB · 2026-04-04
> >
> > Thank you to the authors for running with a range of random seeds and reporting the confidence intervals.  These results do look satisfactory.

---

### Official Review · Reviewer_kovG · 2026-03-13

**Soundness:** 2
**Presentation:** 3
**Significance:** 2
**Originality:** 4
**Overall Recommendation:** 5
**Confidence:** 4

**Summary:**

The paper introduces a Phase-Type decoder to VAE, which admits close-form conditional likelihood while being much more flexible than the usual distribution family used in VAEs. The paper claims this flexibility allows PH-VAE to better handle heavy-tailed distributions.

**Compliance With Llm Reviewing Policy:**

Affirmed.

**Final Justification:**

The paper's novelty was recognized, but there obvious edges. Now they are largely cleared.

**Key Questions For Authors:**

- Table 2 is on Pareto data, which is 1D, so how does Tail CoExErr@99 (a multivariate metric) work here?

Despite the weaknesses listed above, the novel idea proposed in this paper has great potential, and I'm willing to revise the score if the issues are addressed.

**Limitations:**

yes

**Strengths And Weaknesses:**

Strength:
Novel decoder idea. Using a latent-conditioned Phase-Type decoder is a novel way to bring structured stochastic-process likelihoods into VAEs, instead of hard-coding Gaussian, Student-t, or EVT families. It is not tied to one fixed heavy-tail family; it can adapt its effective shape over the observed range while keeping the decoder likelihood analytically manageable.

Weakness 1:
PH-VAE is built on finite Phase-Type distributions, which are asymptotically light-tailed; thus, the model cannot recover the true tail class of Pareto- or regularly-varying data. The appendix acknowledges this and reframes the goal as a finite-range approximation of tail probabilities and extreme quantiles. However, the main paper still claims recovery of “diverse heavy-tailed distributions” and even “power-law dynamics,” which overstates what the method actually guarantees. This limitation should be highlighted more prominently and tested directly through extrapolation or tail-index experiments.

Please clarify the intended scope of the claim and revise the framing accordingly: is PH-VAE meant to model true asymptotic heavy tails, or only heavy-tail-like behaviour over finite observed ranges? If the latter, please revise the title, abstract and state this explicitly in the main text to prevent misleading readers. Furthermore, please provide evidence on how far into the tail the approximation remains reliable.

Weakness 2. The real-data evaluation is substantially less rigorous than the synthetic evaluation. For the Danish fire losses and Google word counts, the paper explicitly relies on qualitative log-log CCDF diagnostics rather than quantitative metrics, so the results mainly show visual agreement with the observed tail, not held-out probability metrics (e.g. ELBO on held-out data) or calibrated extreme-quantile estimation.

---

> ### Author Rebuttal · Authors · 2026-03-30
>
> We sincerely thank the reviewer for their thoughtful, detailed, and constructive feedback. We greatly appreciate the positive assessment of the novelty and potential impact of the Phase-Type decoder. The comments helped us substantially improve the clarity and positioning of the paper. We address each remaining concern below.
>
>
> ### 1. Scope of Heavy-Tail Modeling (Asymptotic vs Finite-Range)
>
> The family of acyclic PH distributions is dense in the set of all probability distributions on R_+ (in the sense of weak convergence). We cite in our paper the corresponding theorem on page 4 as "Thm. 4.2 on page 84 of (Asmussen, 2003)" where also the proof is given. This guarantees, from a theoretical point of view, that any distribution on $\mathbb{R}_+$ can be approximated arbitrarily well by an acyclic PH distribution.
>
> As pointed out by Reviewers kovG and RVRV, and stated also in our paper on page 3, PH distributions are asymptotically light-tailed (they have exponential tail decay). It was however shown in a number of papers that, by appropriate combination of exponential scales, acyclic PH distributions are able to capture different types of heavy tailed behaviors over any finite interval. The seminal paper in this direction is (Feldmann et al., 1998) where proposed technique is limited to distributions with monotone decreasing probability density function. In our paper we cited also (Horvath & Telek, 2000) where this limitation is removed. Both techniques allow us to set an upper limit up to which the PH density follows a given heavy-tailed behavior. An illustration is given in Appendix C.6. A further discussion about "Asymptotic vs. Practical Tail Behavior" is provided in Appendix C.7.
>
> We have revised the title, abstract, and main text to explicitly reflect this finite-range approximation perspective and avoid any ambiguity.
>
> New proposed title: "PH-VAE: Phase-Type Variational Autoencoders for Finite-Range Approximation of Heavy-Tailed Distributions"
>
>
> **Empirical evidence (tail range reliability).**
> To directly address this point, we added truncated-data experiments where models are trained without access to extreme values and evaluated beyond the training range.
>
> | q     | True | PH samples   | Gauss samples | Trunc data | PH err | Gauss err | Trunc err |
> |------|------|------|-------|-------|--------|-----------|-----------|
> | 0.900 | 2.61 | 2.47 | 2.07  | 2.49  | 0.05   | 0.21      | 0.04      |
> | 0.980 | 5.10 | 4.28 | 3.04  | 4.22  | 0.16   | 0.40      | 0.17      |
> | 0.990 | 6.81 | 4.90 | 3.44  | 4.99  | 0.28   | 0.50      | 0.27      |
> | 0.995 | 9.09 | 5.49 | 3.83  | 5.67  | 0.40   | 0.58      | 0.38      |
> | 0.999 |17.78 | 6.76 | 4.56  | 6.40  | 0.62   | 0.74      | 0.64      |
>
> PH-VAE consistently extrapolates better than baseline, although performance degrades further into the tail (q@0.999), as expected. This demonstrates that PH-VAE provides meaningful but controlled extrapolation, consistent with its finite-range modeling objective.
>
>
> ### 2. Quantitative Evaluation on Real Data
>
> Indeed, We fully agree with the reviewer that the real-data evaluation was too qualitative.
>
> In response, we added a **quantitative tail-evaluation protocol** including:
> - tail quantile errors
> - Conditional Value at Risk (CVaR) errors
>
> For example, on the Danish fire dataset:
>
> **Quantile Errors metrics:**
>
> | q     | True | PH   | VAE  | PH rel err | VAE rel err |
> |------|------|------|------|------------|-------------|
> | 0.900 | 5.54 | 8.05 | 5.67 | 0.45       | 0.02        |
> | 0.950 | 9.97 | 13.67| 8.07 | 0.37       | 0.19        |
> | 0.980 | 18.60| 31.71|11.06 | 0.70       | 0.41        |
> | 0.990 | 26.04| 47.64|13.90 | 0.83       | 0.47        |
> | 0.995 | 34.82| 65.49|17.04 | 0.88       | 0.51        |
> | 0.999 |131.55|109.68|23.56 | 0.17       | 0.82        |
>
> **CVaR results:**
>
> | q     | True | PH   | VAE  | PH rel err | VAE rel err |
> |------|------|------|------|------------|-------------|
> | 0.950 | 24.08| 34.92| 11.83| 0.45       | 0.51        |
> | 0.990 | 58.59| 72.39| 18.60| 0.24       | 0.68        |
> | 0.995 | 87.59| 89.60| 22.11| 0.02       | 0.75        |
>
> These results show that classical VAE severely underestimates the tail of the observed distribution of data, while PH-VAE remains significantly closer to the true extreme behavior, particularly for high quantiles and tail-risk measures.
>
>
> ### 3. Clarification on Tail CoExErr@99 (Table 2)
>
> In the 1D setting (e.g., Pareto experiments), the metric reported as Tail CoExErr 99 was in fact a typo and should correspond to Q99 error (quantile error at the 0.99 level), which was corrected.
>
>
>
> Finally, we again sincerely thank the reviewer for his important remarks. As suggested, we have revised the paper to clearly position PH-VAE as a finite-range heavy-tail modeling framework, added quantitative metrics to real data evaluation and corrected the typo in table column name.

---

> > ### Author Rebuttal · Reviewer_kovG · 2026-04-02
> >
> > The new experiments and results are helpful.

---

### Official Review · Reviewer_1r38 · 2026-03-13

**Soundness:** 4
**Presentation:** 3
**Significance:** 4
**Originality:** 3
**Overall Recommendation:** 4
**Confidence:** 4

**Summary:**

This paper proposes PH-VAE, a variational autoencoder for positive-valued heavy-tailed data in which the usual decoder likelihood is replaced by a latent-conditioned Phase-Type (PH) distribution. the model keeps a standard Gaussian latent variable and encoder, but for each output dimension the decoder produces a PH representation in series canonical form, with parameters conditioned on the latent sample. The resulting likelihood factorizes across dimensions given the latent variable, while cross-dimensional dependence is induced through the shared latent representation. Empirically, the paper evaluates PH-VAE on synthetic univariate heavy-tailed distributions (Weibull, Pareto, Lognormal, Burr), real univariate benchmarks such as Danish fire insurance losses and word-frequency counts, and multivariate data including synthetic Student-t-copula constructions and real financial returns. The reported results suggest that PH-VAE substantially improves tail fidelity and extreme-quantile accuracy over Gaussian VAE and also outperforms fixed-family heavy-tail decoders such as t-VAE and xVAE in several settings.

**Compliance With Llm Reviewing Policy:**

Affirmed.

**Key Questions For Authors:**

What are the practical failure modes of PH-VAE?

**Strengths And Weaknesses:**

pros:
1. the paper tackles a real modeling gap rather than an artificial benchmark issue
2. the use of Phase-Type distributions is genuinely original in this context
3. the empirical results on 1D synthetic data are strong and easy to interpret

cons:
1.the comparison to baselines is limited. The main baselines are Gaussian VAE, t-VAE, and xVAE. These are relevant, but the heavy-tail generative modeling literature is broader than this, especially if one allows mixture decoders, flow-based likelihoods, or simpler positive-support alternatives such as Gamma/Lognormal mixtures. In addition, several papers also focus on similar topics like motivating from VAE ELBO and focus on more real dataset, like VIR [1].

2. the real-data evaluation is somewhat too qualitative.
For the Danish fire insurance and word-frequency datasets, the evidence is mainly based on log-log CCDF plots. In addition, is this possible to use some other datasets like AgeDB ?

[1] Variational Imbalanced Regression: Fair Uncertainty Quantification via Probabilistic Smoothing, NeurIPS 2023

---

> ### Author Rebuttal · Authors · 2026-03-30
>
> We sincerely thank the reviewer for their thoughtful, detailed, and highly constructive feedback. We greatly appreciate the positive assessment of the paper’s motivation, technical soundness, and empirical clarity. The suggestions were extremely valuable and helped us significantly strengthen the submission. We address each remaining concern below.
>
>
> ### 1. Baseline Comparisons
>
> We fully agree that the initial set of baselines was not sufficiently broad.
>
> Following the reviewer’s suggestion, We have extended the evaluation to include additional positive-support likelihoods (Lognormal and Gamma decoders). PH-VAE consistently achieves lower error in extreme quantiles and tail KS, while remaining competitive in moderate regimes, confirming that the observed gains are robust across stronger baselines.
>
> | Dataset   | KS_tail (Gamma-VAE) | KS_tail (LogNormal-VAE) | KS_tail (PH-VAE) | Q99_err (Gamma-VAE) | Q99_err (LogNormal-VAE) | Q99_err (PH-VAE) |
> |-----------|--------------------|-------------------------|------------------|---------------------|--------------------------|------------------|
> | lognormal | 0.136 ± 0.048      | **0.026 ± 0.010**       | 0.039 ± 0.013    | 0.306 ± 0.050       | 0.082 ± 0.051            | **0.066 ± 0.071**|
> | pareto    | 0.236 ± 0.100      | 0.072 ± 0.046           | **0.090 ± 0.030**| 0.342 ± 0.030       | 0.197 ± 0.079            | **0.086 ± 0.032**|
> | weibull   | 0.056 ± 0.004      | 0.326 ± 0.023           | **0.019 ± 0.004**| 0.082 ± 0.012       | 2.088 ± 0.381            | **0.025 ± 0.016**|
>
>
>
> ### 2. Quantitative Evaluation on Real Data
>
> We thank the reviewer for pointing out that the original real-data evaluation was too qualitative.
>
> In response, we added a quantitative tail-evaluation protocol including:
> - tail quantile errors (absolute and relative)
> - CVaR errors (absolute and relative)
> For example, in the case of the Danish data:
>
> **Quantile results:**
>
> | q     | True  | PH    | VAE   | PH rel err | VAE rel err |
> |------|-------|-------|-------|------------|-------------|
> | 0.900 | 5.54  | 7.04  | 3.97  | 0.27       | 0.28        |
> | 0.950 | 9.97  | 10.58 | 5.14  | 0.06       | 0.48        |
> | 0.980 | 18.60 | 21.32 | 6.42  | 0.15       | 0.66        |
> | 0.990 | 26.04 | 33.44 | 7.44  | 0.28       | 0.71        |
> | 0.995 | 34.82 | 44.47 | 8.42  | 0.28       | 0.76        |
> | 0.999 |131.55 | 71.33 |10.38  | 0.46       | 0.92        |
>
> **CVaR results:**
>
> | q     | True | PH   | VAE  | PH rel err | VAE rel err |
> |------|------|------|------|------------|-------------|
> | 0.950 | 24.08| 34.92| 11.83| 0.45       | 0.51        |
> | 0.990 | 58.59| 72.39| 18.60| 0.24       | 0.68        |
> | 0.995 | 87.59| 89.60| 22.11| 0.02       | 0.75        |
>
> These results show that Gaussian VAE severely underestimates the tail, while PH-VAE remains much closer to the true extreme behavior, especially for the highest quantiles and CVaR.
>
>
> ### 3. Failure Modes of PH-VAE
>
> We thank the reviewer for this important question and conducted a dedicated analysis.
>
> **(A) Truncated-data extrapolation**
>
> | q     | True | PH   | Gauss | Trunc | PH err | Gauss err | Trunc err |
> |------|------|------|-------|-------|--------|-----------|-----------|
> | 0.900 | 2.61 | 2.47 | 2.07  | 2.49  | 0.05   | 0.21      | 0.04      |
> | 0.980 | 5.10 | 4.28 | 3.04  | 4.22  | 0.16   | 0.40      | 0.17      |
> | 0.990 | 6.81 | 4.90 | 3.44  | 4.99  | 0.28   | 0.50      | 0.27      |
> | 0.995 | 9.09 | 5.49 | 3.83  | 5.67  | 0.40   | 0.58      | 0.38      |
> | 0.999 |17.78 | 6.76 | 4.56  | 6.40  | 0.62   | 0.74      | 0.64      |
>
> PH-VAE still extrapolates better than Gaussian VAE, but performance degrades without tail observations.
> **Failure mode #1:** limited tail exposure => reduced extrapolation accuracy.
>
> **(B) Data regime sensitivity**
>
> | N_train | train time (s) | q95 err | q99 err | q995 err | q999 err |
> |--------|----------------|--------|--------|---------|---------|
> | 500    | 3.17           | 0.09   | 0.04   | 0.15    | 0.40    |
> | 2000   | 11.29          | 0.04   | 0.03   | 0.10    | 0.33    |
> | 5000   | 29.81          | 0.02   | 0.01   | 0.08    | 0.32    |
> | 20000  | 117.77         | 0.02   | 0.08   | 0.02    | 0.27    |
>
> **Failure mode #2:** insufficient data => unstable tail estimates.
>
>
> **(C) High-dimensional setting**
>
> | dim | latent | corr err | q90 err | q95 err | q99 err |
> |-----|--------|----------|--------|--------|--------|
> | 20  | 10     | 4.42     | 0.08   | 0.10   | 0.31   |
> | 50  | 10     | 8.67     | 0.07   | 0.13   | 0.34   |
> | 500 | 20     | 92.48    | 1.17   | 1.27   | 1.93   |
>
> **Failure mode #3:** high dimensionality with limited latent capacity.
>
> These results clarify the operating regimes of PH-VAE, showing that it remains superior to baselines while revealing well-understood and addressable limitations.
>
> We thank the reviewer for the feedback, which helped improve the paper. We hope the additional experiments and clarifications address the concerns and better highlight our contribution.

---

> > ### Author Rebuttal · Reviewer_1r38 · 2026-04-03
> >
> > Thanks for the authors rebuttal

---

### Decision · Program_Chairs · 2026-04-30

**Decision:**

Accept (regular)

**Comment:**

Two reviewers recommend Accept, and two reviewers recommend Weak Accept. Concerns raised by the reviewers, mostly related to empirical results and further validation, were addressed strongly by the authors in the rebuttal. Therefore, I recommend Accept.